# The Effect of Neural Net Architecture on Gradient Confusion & Training Performance

## Abstract

The goal of this paper is to study why typical neural networks train so fast, and how neural network architecture affects the speed of training. We introduce a simple concept called *gradient confusion* to help formally analyze this. When confusion is high, stochastic gradients produced by different data samples may be negatively correlated, slowing down convergence. But when gradient confusion is low, data samples interact harmoniously, and training proceeds quickly. Through novel theoretical and experimental results, we show how the neural net architecture affects gradient confusion, and thus the efficiency of training. We show that increasing the *width* of neural networks leads to *lower* gradient confusion, and thus easier model training. On the other hand, increasing the *depth* of neural networks has the opposite effect. Finally, we observe empirically that techniques like batch normalization and skip connections reduce gradient confusion, which helps reduce the training burden of very deep networks.

## 1 Introduction

Stochastic gradient descent (SGD) (Robbins & Monro, 1951) and its variants with momentum (Sutskever et al., 2013) have become the standard optimization routine for neural networks due to their fast convergence and good generalization properties (Wilson et al., 2017; Keskar & Socher, 2017; Sutskever et al., 2013). Yet the behavior of SGD on high-dimensional neural network models still eludes full theoretical understanding, both in terms of its convergence and generalization properties. In this paper, we study why SGD is so efficient at converging to low loss values on most standard neural networks, and how neural net architecture design affects training performance.

Classical stochastic optimization theory predicts that the learning rate of SGD needs to decrease over time for convergence to be guaranteed to the minimizer of a convex function (Shamir & Zhang, 2013; Bertsekas, 2011). For strongly convex functions for example, such results show that a decreasing learning rate schedule of $O(1/k)$ is required to guarantee convergence to within $\epsilon$-accuracy of the minimizer in $O(1/\epsilon)$ iterations, where $k$ denotes the iteration number. Such decay schemes, however, typically lead to poor performance on standard neural network problems. Neural networks operate in a regime where the number of parameters is much larger than the number of training data. In this regime, SGD seems to converge quickly with constant learning rates. Most neural net practitioners use a constant learning rate for the majority of training, with exponentially decaying learning rate schedules at the end, without seeing the method stall (Krizhevsky et al., 2012; Simonyan & Zisserman, 2014; He et al., 2016; Zagoruyko & Komodakis, 2016). With constant learning rates, theoretical guarantees show that SGD converges quickly to a neighborhood of the minimizer, but then reaches a *noise floor* beyond which it stops converging; this noise floor depends on the learning rate and the variance of the gradients (Moulines & Bach, 2011; Needell et al., 2014). Some more recent results have shown that when models can fit the data completely while being strongly convex, convergence without a noise floor is possible without decaying the learning rate (Schmidt & Roux, 2013; Ma et al., 2017; Bassily et al., 2018; Vaswani et al., 2018).

While these results do give important insights, they do not fully explain the dynamics of SGD on neural nets, and how they relate to overparameterization. Training performance is also highly affected by the neural network architecture. It is common knowledge among neural network practitioners that deeper networks train slower (Bengio et al., 1994; Glorot & Bengio, 2010). This has led to several innovations over the years to get deeper networks to train more easily, such as careful initialization strategies (Glorot & Bengio, 2010; He et al., 2015; Zhang et al., 2019), residual connections (He et al.,

2016), and various normalization schemes like batch normalization (Ioffe & Szegedy, 2015) and weight normalization (Salimans & Kingma, 2016). Furthermore, there is ample evidence to indicate that wider networks are easier to train (Zagoruyko & Komodakis, 2016; Nguyen & Hein, 2017; Lee et al., 2019), and recent theoretical results suggest that the dynamics of SGD simplify considerably for very wide networks (Jacot et al., 2018; Lee et al., 2019). Several prior works have investigated the difficulties of training deep networks (Glorot & Bengio, 2010; Balduzzi et al., 2017), and the benefits of width (Nguyen & Hein, 2017; Lee et al., 2019; Du et al., 2018; Allen-Zhu et al., 2018). This work advances the existing literature by identifying and analyzing a condition that enables us to theoretically and empirically establish novel direct relationships between layer width, network depth, problem dimensionality, and SGD dynamics on overparameterized networks.

**Our contributions.** Typical neural nets are *overparameterized* (*i.e.,* the number of parameters exceed the number of training points). In this paper, we ask how this overparameterization, and more specifically the architecture of a neural net, affects the dynamics of SGD. We answer this question through extensive theoretical and experimental studies and show how network width, depth, batch normalization and skip connections affect the dynamics. We emphasize that our main contributions are *conceptual*. In particular, following are our main contributions.[1]

- We identify a condition, termed *gradient confusion*, that impacts the convergence properties of SGD on overparameterized models. We prove that high gradient confusion may lead to slower convergence, while convergence is accelerated (and could be faster than predicted by existing theory) if confusion is low indicating a regime where constant learning rates work well in practice (sections 2 and 3). We use this gradient confusion condition as the main proxy, to study the effect of various architecture choices on convergence.

- We study the effect of neural net architecture on gradient confusion (section 4), and prove (a) gradient confusion increases as the network depth increases, and (b) at initialization, wider networks have lower gradient confusion. This indicates that deeper networks are more difficult to train and wider networks improves trainability of neural nets. Directly analyzing the gradient confusion bound enables us to derive *novel and tight results on the direct effect of depth and width*, without requiring arguably restrictive assumptions like infinitely wide networks (Schoenholz et al., 2016; Lee et al., 2019). Our results hold for a large family of neural networks with non-linear activations and a large class of loss-functions.

- We test our theoretical predictions using extensive experiments on wide residual networks (WRNs) (Zagoruyko & Komodakis, 2016), convolutional networks (CNNs) and multi-layer perceptrons (MLPs) for image classification tasks on CIFAR-10, CIFAR-100 and MNIST (section 5 and appendix A). We find that our theoretical results consistently hold across all our experiments. We further show that innovations like batch normalization and skip connections in residual networks help lower gradient confusion, thus indicating why standard neural networks that employ such techniques are so efficiently trained using SGD.

## 2 GRADIENT CONFUSION

**Notations.** We denote vectors in bold lower-case and matrices in bold upper-case. We use $(\mathbf{W})_{i,j}$ to indicate the $(i, j)$ cell in matrix $\mathbf{W}$ and $(\mathbf{W})_i$ for the $i^{\text{th}}$ row of matrix $\mathbf{W}$. $\|\mathbf{W}\|$ denotes the operator norm of $\mathbf{W}$. $[N]$ denotes $\{1, 2, \ldots, N\}$ and $[N]_0$ denotes $\{0, 1, \ldots, N\}$.

**Preliminaries.** Given $N$ training points (specified by the corresponding loss functions $\{f_i\}_{i\in[N]}$), we use SGD to solve empirical risk minimization problems of the form,

$$\min_{\mathbf{w}\in\mathbb{R}^d} F(\mathbf{w}) := \min_{\mathbf{w}\in\mathbb{R}^d} \tfrac{1}{N} \sum_{i=1}^{N} f_i(\mathbf{w}), \tag{1}$$

using the following iterative update rule for $T$ rounds:

$$\mathbf{w}_{k+1} = \mathbf{w}_k - \alpha\nabla\tilde{f}_k(\mathbf{w}_k). \tag{2}$$

Here $\alpha$ is the learning rate and $\tilde{f}_k$ is a function chosen uniformly at random from $\{f_i\}_{i\in[N]}$ at iteration $k \in [T]$. We use $\mathbf{w}^\star$ to denote the optimal solution, i.e., $\mathbf{w}^\star = \arg\min_{\mathbf{w}} F(\mathbf{w})$.

---

[1]Due to space constraints, all proofs and several additional experiments are delegated to the appendix.

**Gradient confusion.** SGD works by iteratively selecting a random function $\tilde{f}_k$, and modifying the parameters to move in the direction of the negative gradient of the objective term $\tilde{f}_k$. It may happen that the selected gradient $\nabla\tilde{f}_k$ is negatively correlated with the gradient of another term $\nabla f_j$. When the gradients of different mini-batches are negatively correlated, the objective terms disagree on which direction the parameters should move, and we say that there is *gradient confusion*.[2]

**Definition 2.1.** *A set of objective functions $\{f_i\}_{i \in [N]}$ has gradient confusion bound $\eta \geq 0$ if the pair-wise inner products between gradients satisfy, for a fixed $\mathbf{w} \in \mathbb{R}^d$,*

$$\langle \nabla f_i(\mathbf{w}), \nabla f_j(\mathbf{w}) \rangle \geq -\eta, \ \forall i \neq j \in [N]. \tag{3}$$

**Remarks.** Note that while the gradient confusion bound $\eta$ is defined for the worst-case gradient inner product, all the results in our paper can be trivially extended to using a bound on the average gradient inner product: $\sum_{i,j=1}^{N} \langle \nabla f_i(\mathbf{w}), \nabla f_j(\mathbf{w}) \rangle / N^2 \geq -\eta$. All theoretical results would remain the same up to constants. Further, note that definition 2.1 is applicable even when the stochastic gradients are averaged over minibatches of size $B$. For minibatches of size $B$, the variance of the gradient inner product scales down as $1/B^2$, and thus $\eta$ is expected to decrease as $B$ grows.

**Observations in simplified settings.** SGD converges fast when gradient confusion is low along its path. To see why, consider the case of training a logistic regression model on a dataset with *orthogonal* vectors. We have $f_i(\mathbf{w}) = \ell(y_i\mathbf{x}_i^\top\mathbf{w})$, where $\ell : \mathbb{R} \to \mathbb{R}$ is the logistic loss, $\{\mathbf{x}_i\}_{i \in [N]}$ is a set of orthogonal training vectors, and $y_i \in \{-1, 1\}$ is the label for the $i^{\text{th}}$ training example. We then have $\nabla f_i(\mathbf{w}) = \zeta_i\mathbf{x}_i$, where $\zeta_i = y_i\ell'(y_i \cdot \mathbf{x}_i^\top\mathbf{w})$. Note that the gradient confusion is 0 since $\langle \nabla f_i(\mathbf{w}), \nabla f_j(\mathbf{w}) \rangle = \zeta_i\zeta_j\langle \mathbf{x}_i, \mathbf{x}_j \rangle = 0, \forall i, j \in [N]$ and $i \neq j$. Thus, an update in the gradient direction $f_i$ has *no* effect on the loss value of $f_j$ for $i \neq j$.

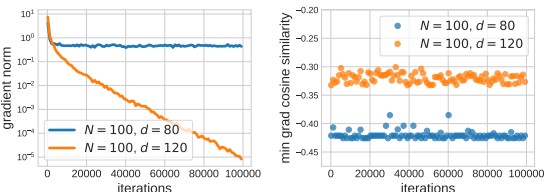

Figure 1: Linear regression on an over-parameterized ($d = 120$) and under-parameterized ($d = 80$) model with $N = 100$ samples generated randomly from a Gaussian, trained using SGD with minibatch size 1. Plots are averaged over 3 independent runs. Gradient cosine similarities were calculated over all pairs of gradients.

In this case, SGD decouples into (deterministic) gradient descent on each objective term separately, and we can expect to see the fast convergence rates attained by gradient descent.

Can we expect a problem to have low gradient confusion in practice? From the logistic regression problem, we have: $|\langle \nabla f_i(\mathbf{w}), \nabla f_j(\mathbf{w}) \rangle| = |\langle \mathbf{x}_i, \mathbf{x}_j \rangle| \cdot |\zeta_i\zeta_j|$. This inner product is expected to be small for all $\mathbf{w}$; the logistic loss satisfies $|\zeta_i\zeta_j| < 1$, and for fixed $N$ the quantity $\max_{ij} |\langle \mathbf{x}_j, \mathbf{x}_i \rangle|$ is $O(1/\sqrt{d})$ whenever $\{\mathbf{x}_i\}$ are randomly sampled from a sphere (see lemma B.1 for the formal statement).[3] Thus, we would expect a random linear model to have nearly orthogonal gradients, when the number of parameters is "large" and the number of training data is "small", i.e., when the model is over-parameterized. This is further evidenced by a toy example in figure 1, where we show a slightly overparameterized linear regression model can have much faster convergence rates, as well as lower gradient confusion, compared to the underparameterized model.

Now consider more general neural net problems. There is evidence that the Hessian at the minimizer is very low rank for many standard overparameterized neural net models (Sagun et al., 2017; Cooper, 2018; Chaudhari et al., 2016; Wu et al., 2017; Ghorbani et al., 2019). What does this imply for the gradient confusion? For clarity in presentation, suppose each $f_i$ has a minimizer at the origin (the same argument can be easily extended to the more general case). Suppose also that there is a Lipschitz constant for the Hessian of each function $f_i$ that satisfies $\|\mathbf{H}_i(\mathbf{w}) - \mathbf{H}_i(\mathbf{w}')\| \leq L_H\|\mathbf{w} - \mathbf{w}'\|$ (note that this is a standard optimization assumption (Nesterov, 2018), with evidence that it is applicable for neural nets (Martens, 2016)). Then $\nabla f_i(\mathbf{w}) = \mathbf{H}_i\mathbf{w} + \mathbf{e}$, where $\mathbf{e}$ is an error term bounded as: $\|\mathbf{e}\| \leq \frac{1}{2}L_H\|\mathbf{w}\|^2$, and we use the shorthand $\mathbf{H}_i$ to denote $\mathbf{H}_i(\mathbf{0})$. Then we have (appendix C):

$$|\langle \nabla f_i(\mathbf{w}), \nabla f_j(\mathbf{w}) \rangle| \leq \|\mathbf{w}\|^2\|\mathbf{H}_i\|\|\mathbf{H}_j\| + \frac{1}{2}L_H\|\mathbf{w}\|^3(\|\mathbf{H}_i\| + \|\mathbf{H}_j\|) + \frac{1}{4}L_H^2\|\mathbf{w}\|^4.$$

---

[2]This is related to both gradient variance and gradient diversity (Yin et al., 2017), but with important differences. See section 6 for more discussion.

[3]Generally, this is true whenever $\mathbf{x}_i = \frac{1}{\sqrt{d}}\mathbf{y}_i$, where $\mathbf{y}_i$ is an isotropic random vector (Vershynin, 2018).

If the Hessians are sufficiently random and low-rank (e.g., of the form $\mathbf{H}_i = \mathbf{a}_i\mathbf{a}_i^\top$ where $\mathbf{a}_i \in \mathbb{R}^{N \times r}$ are randomly sampled from a unit sphere), then one would expect the terms in this expression to be small for all $\mathbf{w}$ within a neighborhood of the minimizer. While a bit non-rigorous, this nonetheless suggests that for many standard neural network models, the gradient confusion might be small for a large class of weights near the minimizer.

The above arguments are rather informal, and ignore issues like the effect of the structure of neural networks. In the following sections, we rigorously analyze the effect of gradient confusion on the speed of convergence on non-convex problems, and the effect of width and depth of the neural net architecture on the gradient confusion.

## 3 SGD is Efficient when Gradient Confusion is Low

Several prior papers have analyzed the convergence rates of constant learning rate SGD (Nedić & Bertsekas, 2001; Moulines & Bach, 2011; Needell et al., 2014; Dieuleveut et al., 2017). These results show that for strongly convex and Lipschitz smooth functions, SGD with a constant learning rate $\alpha$ converges *linearly* to a neighborhood of the minimizer. The noise floor it converges to depends on the learning rate $\alpha$ and the variance of the gradients at the minimizer, i.e., $\mathbb{E}_i\|\nabla f_i(\mathbf{w}^\star)\|^2$. To guarantee convergence to $\epsilon$-accuracy in such a setting, the learning rate needs to be small, i.e., $\alpha = O(\epsilon)$, and the method requires $T = O(1/\epsilon)$ iterations. Some more recent results show convergence of constant learning rate SGD without a noise floor and without small step sizes for models that can completely fit the data (Schmidt & Roux, 2013; Ma et al., 2017; Bassily et al., 2018; Vaswani et al., 2018).

The gradient confusion bound is related to these classical results. Cauchy-Swartz inequality implies that if $\mathbb{E}_i\|\nabla f_i(\mathbf{w}^\star)\|^2 = O(\epsilon)$, then $\mathbb{E}_{i,j}|\langle\nabla f_i(\mathbf{w}^\star), \nabla f_j(\mathbf{w}^\star)\rangle| = O(\epsilon), \forall i, j$. Thus the gradient confusion at the minimizer is small when the variance of the gradients at the minimizer is small. Further note that when the variance of the gradients at the minimizer is $O(\epsilon)$, a direct application of the results in (Moulines & Bach, 2011; Needell et al., 2014) shows that constant learning rate SGD has fast convergence to $\epsilon$-accuracy in $T = O(\log(1/\epsilon))$ iterations, without the learning rate needing to be small. Generally however, bounded gradient confusion does not provide a bound on the variance of the gradients (see section 6 for more discussion). Thus, it is instructive to derive convergence bounds of SGD explicitly in terms of the gradient confusion to properly understand its effect.

We begin by considering functions satisfying the Polyak-Lojasiewicz (PL) inequality (Lojasiewicz, 1965), a condition related to, but weaker than, strong convexity, and provide bounds on the rate of convergence in terms of the optimality gap. Then we look at a broader class of smooth non-convex functions, and analyze convergence to a stationary point. We first make two standard assumptions.

**(A1)** $\{f_i\}_{i \in [N]}$ are *Lipschitz smooth*: $f_i(\mathbf{w}') \leq f_i(\mathbf{w}) + \nabla f_i(\mathbf{w})^\top(\mathbf{w}' - \mathbf{w}) + \frac{L}{2}\|\mathbf{w}' - \mathbf{w}\|^2$.

**(A2)** $\{f_i\}_{i \in [N]}$ satisfy the *PL inequality*: $\frac{1}{2}\|\nabla f_i(\mathbf{w})\|^2 \geq \mu(f_i(\mathbf{w}) - f_i^\star), f_i^\star = \min_{\mathbf{w}} f_i(\mathbf{w})$.

**Theorem 3.1.** *If the objective function satisfies (A1) and (A2), and has gradient confusion $\eta$, SGD with updates of the form* (2) *converges linearly to a neighborhood of the minima of problem* (1) *as:*

$$\mathbb{E}[F(\mathbf{w}_T) - F^\star] \leq \rho^T(F(\mathbf{w}_0) - F^\star) + \frac{\alpha\eta}{1-\rho},$$

*where $\alpha < \frac{2}{NL}$, $\rho = 1 - \frac{2\mu}{N}\left(\alpha - \frac{NL\alpha^2}{2}\right)$, $F^\star = \min_{\mathbf{w}} F(\mathbf{w})$ and $\mathbf{w}_0$ is the initialized weights.*

This result shows that SGD converges *linearly* to a neighborhood of a minimizer, and the size of this neighborhood depends on the level of gradient confusion. When the gradient confusion is small, i.e., $\eta = O(\epsilon)$, SGD has fast convergence to $O(\epsilon)$-accuracy in $T = O(\log(1/\epsilon))$ iterations, without requiring the learning rate to be vanishingly small. We now extend this to general smooth functions.

**Theorem 3.2.** *If the objective satisfies (A1) and has gradient confusion bound $\eta$, then SGD converges to a neighborhood of a stationary point as:*

$$\min_{k=1,\ldots,T} \mathbb{E}\|\nabla F(\mathbf{w}_k)\|^2 \leq \frac{\rho(F(\mathbf{w}_1) - F^\star)}{T} + \rho\eta,$$

*for learning rate $\alpha < \frac{2}{NL}$, $\rho = \frac{2N}{2-NL\alpha}$, and $F^\star = \min_{\mathbf{w}} F(\mathbf{w})$.*

Thus, as long as $\eta = O(1/T)$, SGD has fast $O(1/T)$ convergence on smooth non-convex functions. Theorems 3.1 and 3.2 predict an initial phase of optimization with fast convergence to the neighborhood of a minimizer or a stationary point. This behavior is often observed when optimizing neural

nets (Darken & Moody, 1992; Sutskever et al., 2013), where a constant learning rate reaches a high level of accuracy on the model. As we show in subsequent sections, this is expected since for neural networks typically used, the gradient confusion is expected to be low. Convergence slows down as the iterates approach the noise floor, and at this point typically practitioners employ exponentially decaying learning rate schedules (Krizhevsky et al., 2012; Simonyan & Zisserman, 2014; He et al., 2016; Zagoruyko & Komodakis, 2016; Ge et al., 2019). See section 6 for more discussion on theorems 3.1 and 3.2, and how they relate to results in previous works. We stress that our goal is not to study convergence rates per se, nor is it to prove state-of-the-art rate bounds for this class of problems. The main intention is to show the direct effect that the gradient confusion bound has on the convergence rate and the noise floor that constant learning rate SGD converges to. As we show in the following sections, this new perspective in terms of the gradient confusion helps us more directly understand how neural net architecture design affects SGD dynamics and why.

## 4    Effect of Neural Net Architecture on Gradient Confusion

To draw a rigorous connection between neural net structure and training performance, we analyze gradient confusion for generic (i.e., random) model problems using methods from high-dimensional probability. In particular, this section considers the following scenarios: (a) Random data drawn from a unit sphere and the weights in a ball around the local minimizer (theorem 4.1 and corollary 4.1). (b) Random weights using standard initialization schemes and both arbitrary bounded data (theorem 4.2, part 1) and random data drawn from a unit sphere (theorem 4.2, part 2). Our results cover a wide range of scenarios compared to prior work (e.g., Chen et al. (2018); Schoenholz et al. (2016); Balduzzi et al. (2017)), require minimal additional assumptions, and hold for a large family of neural nets with non-linear activations and a large class of loss-functions. In particular, our results hold for fully connected networks (and convolutional networks in some cases) with the square-loss and logistic-loss functions, and commonly used non-linear activations such as sigmoid, tanh and ReLU.

**(a) Random Data, Bounded Weights Around Minimizer.** In this subsection, we consider training data of the form $\mathcal{D} = \{(\mathbf{x}_i, \mathcal{C}(\mathbf{x}_i))\}_{i \in [N]}$, for some labeling function $\mathcal{C} : \mathbb{R}^d \to [-1, 1]$, and with data points $\{\mathbf{x}_i\}$ drawn uniformly from the surface of a $d$-dimensional unit sphere. The labeling function satisfies $|\mathcal{C}(\mathbf{x})| \le 1$ and $\|\nabla_{\mathbf{x}} \mathcal{C}(\mathbf{x})\|_2 \le 1$ for $\|\mathbf{x}\| \le 1$. Note that this automatically holds for every model considered in this paper where the labeling function is *realizable* (i.e., where the model can express the labeling function using its parameters). More generally, this assumes a Lipschitz condition on the labels (i.e., the labels don't change too quickly with the inputs). In this paper, we consider two loss-functions, namely, square-loss for regression and logistic loss function for classification. The square-loss function is defined as $f_i(\mathbf{w}) = \frac{1}{2}(\mathcal{C}(\mathbf{x}_i) - g_{\mathbf{w}}(\mathbf{x}_i))^2$ and the logistic function is defined as $f_i(\mathbf{w}) = \log(1 + \exp(-\mathcal{C}(\mathbf{x}_i) g_{\mathbf{w}}(\mathbf{x}_i)))$. Here, $g_{\mathbf{w}} : \mathbb{R}^d \to \mathbb{R}$ denotes the parameterized function we fit to the training data and $f_i(\mathbf{w})$ denotes the loss-function of hypothesis $g_{\mathbf{w}}$ on data point $\mathbf{x}_i$.

Formally, let $\mathbf{W}_0 \in \mathbb{R}^{\ell_1 \times d}$ and $\{\mathbf{W}_p\}_{p \in [\beta]}$ such that $\mathbf{W}_p \in \mathbb{R}^{\ell_p \times \ell_{p-1}}$ be the given weight matrices. Let $\mathbf{W}$ denote the tuple $(\mathbf{W}_p)_{p \in [\beta]_0}$. Define $\ell := \max_{p \in [\beta]} \ell_p$ to be the *width* and $\beta$ to be the *depth* of the neural network. Then, the model $g_{\mathbf{W}}$ is defined as

$$g_{\mathbf{W}}(\mathbf{x}) := \sigma(\mathbf{W}_\beta \sigma(\mathbf{W}_{\beta-1} \dots \sigma(\mathbf{W}_1 \sigma(\mathbf{W}_0 \mathbf{x})) \dots)), \tag{4}$$

where $\sigma$ denotes the non-linear activation function applied point-wise to its arguments. We assume that the non-linear activation is given by a function $\sigma(x)$ with the following properties.

- **(P1) Boundedness:** $|\sigma(x)| \le 1$ for vector $x \in [-1, 1]$.

- **(P2) Bounded differentials:** Let $\sigma'(x)$ and $\sigma''(x)$ denote the first and second sub-differentials respectively. Then, $|\sigma'(x)| \le 1$ and $|\sigma''(x)| \le 1$ for all $x \in [-1, 1]$.

When $\|\mathbf{x}\| \le 1$, as in our random data model, activation functions such as *sigmoid*, *tanh*, *softmax* and *ReLU* satisfy these requirements. Additionally, we make the following assumption on the weights.

**Assumption 1** (Small Weights). *We assume that the operator norm of the weight matrices $\{\mathbf{W}_i\}_{i \in [\beta]_0}$ are bounded above by 1. In other words, for every $i \in [\beta]_0$ we have $\|\mathbf{W}_i\| \le 1$.*

The operator norm of the weight matrices $\|\mathbf{W}\|$ being close to 1 is important for the trainability of neural nets, as it ensures that the input signal is passed through the net without exploding or shrinking

across layers (Glorot & Bengio, 2010). Proving non-vacuous bounds in case of such blow-ups in magnitude of the signal or the gradient is not possible in general, and thus, we consider this restricted class of weights. The small-weights assumption is not just a theoretical concern, but also usually enforced in practice using *weight decay* regularizers of the form $\sum_i \|W_i\|_F^2$, which keep the weights small during optimization. See appendix F for further discussion on the small weights assumption.

We now prove concentration bounds for the gradient confusion on neural nets.

**Theorem 4.1.** *Consider the problem of training neural nets (equation 4) using either the square-loss or the logistic-loss function. Let $\eta > 0$ be a given constant. Let the weights satisfy assumption 1 and the non-linearities in each layer satisfy properties (P1) and (P2). For some fixed constant $c > 0$, the gradient confusion bound in equation 3 holds with probability at least*

$$1 - N^2 \exp\left(\frac{-cd\eta^2}{16\zeta_0^4(\beta+2)^4}\right),$$

*For both the square-loss and the logistic-loss functions, $\zeta_0 \leq 2\sqrt{\beta}$ (from lemma D.1).*

Thus, theorem 4.1 shows that, for a given dimension $d$ and number of samples $N$, when the network depth $\beta$ decreases, the probability that the gradient confusion bound in equation 3 holds increases, and vice versa. Note that the convergence rate results of SGD in section 3 assume that the gradient confusion bound holds at every point along the path of SGD. On the other hand, theorem 4.1 shows concentration bounds for the gradient confusion at a fixed weight $\mathbf{W}$. Thus, to ensure that the above result is relevant for the convergence of SGD on overparameterized models, we now make the concentration bound in theorem 4.1 *uniform over all weights inside a ball $\mathcal{B}_r$ of radius $r$*.

**Corollary 4.1** (Uniform concentration for all weights around the minimizer)**.** *Select a point $\mathbf{W} = (\mathbf{W}_0, \mathbf{W}_1, \ldots, \mathbf{W}_\beta)$, satisfying assumption 1. Consider a ball $\mathcal{B}_r$ centered at $\mathbf{W}$ of radius $r > 0$. If the data $\{\mathbf{x}_i\}_{i\in[N]}$ are sampled uniformly from a unit sphere, then the gradient confusion bound in equation 3 holds uniformly at all points $\mathbf{W}' \in \mathcal{B}_r$ with probability at least*

$$1 - N^2 \exp\left(-\frac{cd\eta^2}{64\zeta_0^4(\beta+2)^4}\right), \qquad \text{if } r \leq \eta/4\zeta_0^2,$$
$$1 - N^2 \exp\left(-\frac{cd\eta^2}{64\zeta_0^4(\beta+2)^4} + \frac{8d\zeta_0^2 r}{\eta}\right), \qquad \text{otherwise.}$$

Thus, corollary 4.1 shows that the probability that the gradient confusion bound holds decreases with increasing depth, for all weights in a ball around the minimizer. This explains why training very deep models is hard and typically slow with SGD (Bengio et al., 1994; Glorot & Bengio, 2010). Note that this is also related to the *shattered gradients* phenomenon (Balduzzi et al., 2017) that arises with depth (see appendix 6 for more discussion). This naturally raises the question why modern deep neural networks are so efficiently trained using SGD. While careful initialization strategies prevent vanishing or exploding gradients making deeper networks trainable, these strategies still suffer from high gradient confusion for very deep networks (as we show below in theorem 4.2). Thus, in section 5, we empirically study how popular techniques like skip connections (He et al., 2016) and batch normalization (Ioffe & Szegedy, 2015) affect gradient confusion. We find that these techniques drastically lower gradient confusion, making very deep networks significantly easier to train. Note that the above results automatically hold for convolutional nets, since a convolution operation on $\mathbf{x}$ can be represented as a matrix multiplication $\mathbf{U}\mathbf{x}$ for an appropriate Toeplitz matrix $\mathbf{U}$.

**(b) Standard Weight Initializations and the Effect of Layer Width.** Note that on assuming $\|\mathbf{W}\| \leq 1$ for each weight matrix $\mathbf{W}$ in our results in part 1 of section 4, the dependence of gradient confusion on the layer width goes away in general. A simple example that illustrates this is to consider the case where each weight matrix in the neural network has exactly one non-zero element, which is set to 1. The operator norm of each such weight matrix is 1, but the forward or backward propagated signals would not depend on the width. Thus, to better understand the effect of the layer width, in this subsection we focus on the behavior of neural nets at initialization by considering standard weight initialization strategies used when training neural nets. For completeness, we consider both the case where the data is arbitrary but bounded, as well as where the data is randomly drawn from a unit sphere. A key point used in the following results is that typical weight initialization techniques ensure that the operator norm is bounded by 1 with high probability, thus enabling us to derive tight bounds on the gradient confusion. We consider the following weight initialization strategy.

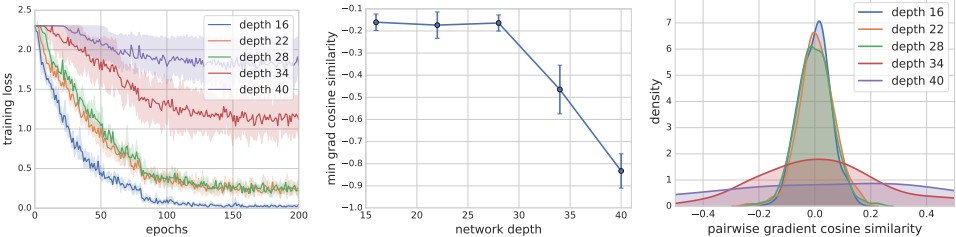

Figure 2: The effect of network depth with CNN-$\beta$-2 on CIFAR-10. *Left plot*: convergence curves of SGD, *Middle plot*: minimum of pairwise gradient cosine similarities at the end of training, *Right plot*: kernel density estimate of the pairwise gradient cosine similarities at the end of training (over all independent runs).

**Strategy 4.1.** $\mathbf{W}_0 \in \mathbb{R}^{\ell \times d}$ *has independent* $\mathcal{N}(0, \frac{1}{d})$ *entries. For every* $p \in [\beta]$, *the weights* $\mathbf{W}_p \in \mathbb{R}^{\ell_p \times \ell_{p-1}}$ *have independent* $\mathcal{N}\left(0, \frac{1}{\kappa \ell_{p-1}}\right)$ *entries for some constant* $\kappa > 0$.

This initialization strategy with different settings of $\kappa$ are used almost universally for neural networks (Glorot & Bengio, 2010; He et al., 2015). The following theorem shows how the width $\ell := \max_{p \in [\beta]} \ell_p$ and the depth $\beta$ affect the gradient confusion condition. In particular, *as width increases or depth decreases the probability that the gradient confusion bound (equation 3) holds increases.*

**Theorem 4.2** (Neural nets with randomly chosen weights). *Let* $\mathbf{W}_0, \mathbf{W}_1, \ldots, \mathbf{W}_\beta$ *be weight matrices chosen according to strategy 4.1. There exists fixed constants* $c_1, c_2 > 0$ *such that we have:*

1. *Consider a fixed but arbitrary dataset* $\mathbf{x}_1, \mathbf{x}_2, \ldots, \mathbf{x}_N$ *with* $\|\mathbf{x}_i\| \le 1$ *for every* $i \in [N]$. *For* $\eta > 4$, *the gradient confusion bound in equation 3 holds with probability at least*

$$1 - \beta \exp\left(-c_1 \kappa^2 \ell^2\right) - N^2 \exp\left(\frac{-c\ell^2 \beta(\eta-4)^2}{64\zeta_0^4(\beta+2)^4}\right).$$

2. *If the dataset* $\{\mathbf{x}_i\}_{i \in [N]}$ *is such that each* $\mathbf{x}_i$ *is an i.i.d. sample from the surface of* $d$-*dimensional unit sphere, then for every* $\eta > 0$ *the gradient confusion bound in equation 3 holds with probability at least*

$$1 - \beta \exp\left(-c_1 \kappa^2 \ell^2\right) - N^2 \exp\left(\frac{-c_2(\ell d + \ell^2 \beta)\eta^2}{16\zeta_0^4(\beta+2)^4}\right).$$

Thus, theorem 4.2 shows that layer width improves the trainability of deep networks under most standard initialization techniques. Note that for reasons described above, almost all recent theoretical results on neural nets have analyzed the effect of width at initialization. Further, one can prove that when the layers are very wide, the weights move very little, and most of the conditions at initialization persist during training, considerably simplifying the learning dynamics of gradient descent (Jacot et al., 2018; Lee et al., 2019). Under this simplifying assumption of very wide networks, several authors have analyzed the effect of width during optimization on very wide networks (Allen-Zhu et al., 2018; Du et al., 2018). Under this assumption and using a standard application of the techniques used in prior work, we can likely show that theorem 4.2 also holds during optimization. In the next section (and in appendix A), we show substantial empirical evidence that, given a sufficiently deep network, increasing the layer width often helps in lowering gradient confusion and speeding up convergence for a range of models, and that these effects persist throughout optimization for most models.

## 5 EXPERIMENTAL RESULTS

To test our theoretical results and to probe why standard neural nets are efficiently trained with SGD, we now present experimental results showing the effect of the neural network architecture on the convergence of SGD and gradient confusion. It is worth noting that theorems 3.1 and 3.2 indicate that we would expect the effect of gradient confusion to be most prominent closer to the end of training.

We performed experiments on wide residual networks (WRNs) (Zagoruyko & Komodakis, 2016), convolutional networks (CNNs) and multi-layer perceptrons (MLPs) for image classification tasks on CIFAR-10, CIFAR-100 and MNIST. We present results for CNNs on CIFAR-10 in this section,

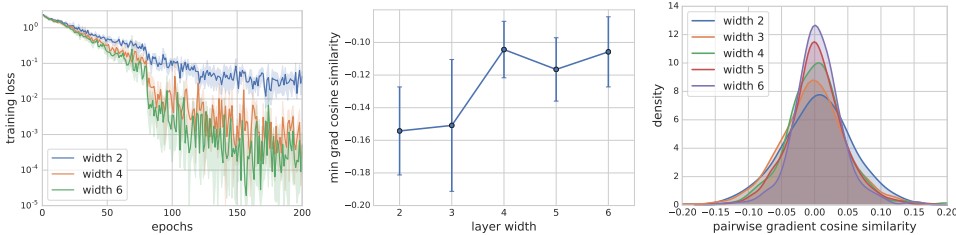

Figure 3: The effect of width with CNN-16-$\ell$ on CIFAR-10. *Left plot*: convergence curves of SGD (for cleaner figures, we plot results for width factors 2, 4 and 6 here), *Middle plot*: minimum of pairwise gradient cosine similarities at the end of training, *Right plot*: kernel density estimate of the pairwise gradient cosine similarities at the end of training (over all independent runs).

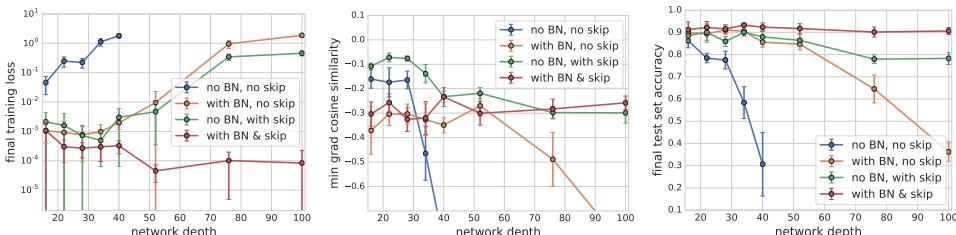

Figure 4: The effect of adding skip connections and batch normalization to CNN-$\beta$-2 on CIFAR-10. Plots show the optimal training loss (*left plot*), minimum pairwise gradient cosine similarities (*middle plot*), and test set accuracies (*right plot*) at the end of training.

and present all other results in appendix A. We use CNN-$\beta$-$\ell$ to denote WRNs that have no skip connections or batch normalization, with a depth $\beta$ and width factor $\ell$.[4] We turned off dropout and weight decay for all our experiments. We used SGD as the optimizer without any momentum. Following Zagoruyko & Komodakis (2016), we ran all experiments for 200 epochs with minibatches of size 128, and reduced the initial learning rate by a factor of 10 at epochs 80 and 160. We used the MSRA initializer (He et al., 2015) for the weights as is standard for this model, and used the same preprocessing steps for the CIFAR-10 images as described in Zagoruyko & Komodakis (2016). We ran each experiment 5 times, and we show the standard deviation across runs in our plots. We tuned the optimal initial learning rate for each model over a logarithmically-spaced grid and selected the run that achieved the lowest training loss value. To measure gradient confusion, at the end of every training epoch, we sampled 100 pairs of mini-batches each of size 128 (the same size as the training batch size). We calculated gradients on each mini-batch, and then computed pairwise cosine similarities. See appendix A.2 for more details on the experimental setup and architectures used.

**Effect of depth.**    To test our theoretical results, we consider CNNs with a fixed width factor of 2 and varying network depth. From figure 2, we see that our theoretical results are backed by the experiments: increasing depth slows down convergence, and increases gradient confusion. We also notice that with increasing depth, the density of pairwise gradient cosine similarities concentrates less sharply around 0 (indicating higher variance), which makes the network harder to train.

**Effect of width.**  We now consider CNNs with a fixed depth of 16 and varying width factors. From figure 3, we see that increasing width results in faster convergence and lower gradient confusion. We further see that gradient cosine similarities concentrate around 0 with growing width, indicating that SGD decouples across the training samples with growing width. Note that the smallest network considered is still overparameterized and achieves a high level of performance (see appendix A.3).

**Effect of batch normalization and skip connections.**    To help understand why many standard deep nets are so efficiently trained using SGD, we test the effect of adding skip connections and batch normalization to CNNs of fixed width and varying depth. Figure 4 shows that adding skip connections or batch normalization individually help in training deeper models, but these models still suffer from worsening results and increasing gradient confusion as the network gets deeper. Both these techniques together keep the gradient confusion relatively low even for very deep networks,

---

[4]The width factor denotes the number of filters relative to the original ResNet model.

significantly improving trainability of deep models. Note that all these observations are consistent with previous work (Balduzzi et al., 2017; Santurkar et al., 2018; Yang et al., 2019).

## 6 CONNECTIONS TO RELATED WORK

**The interpolation condition and connections to the gradient variance**: The gradient confusion condition has interesting connections to the variance of the gradients. If we assume bounded gradient variance $\sigma^2$, we can bound the gradient confusion parameter $\eta$ in terms of other quantities. For example, suppose the true gradient $g$ is defined as $g = g_1/2 + g_2/2$. Then we have:

$$|\langle g_1, g_2 \rangle| \le \|g_1\|\|g_2\| \le 1/2 \cdot (\|g_1\|^2 + \|g_2\|^2) = 1/2 \cdot (\|g_1 - g\|^2 + \|g_2 - g\|^2) + \|g\|^2 = \sigma^2 + \|g\|^2,$$

where all gradients are defined at the same $\mathbf{w}$.

More interestingly however, it is not possible in general to bound the gradient variance in terms of the gradient confusion parameter. As a counter-example, consider a problem with the following distribution on the stochastic gradients: $\frac{1}{1-p}$ samples with gradient $\frac{1}{\epsilon}$ and $\frac{1}{p}$ samples with gradient $\epsilon$, where $p = \epsilon \to 0$. In this case, the gradients are positive, so gradient confusion $\eta = 0$. The mean of the gradients is given by $1 + \epsilon(1 - \epsilon)$, which remains bounded as $\epsilon \to 0$. On the other hand, the variance (and thus the squared norm of the stochastic gradients) is actually unbounded ($O(1/\epsilon)$ as $\epsilon \to 0$). While this is a contrived example, it does show why such a bound does not exist.

A consequence of this is that in theorems 3.1 and 3.2 in section 3, the "noise term" (i.e., the second term in the RHS of the convergence bounds) does not depend on the learning rate in the general case. If gradients have unbounded variance, lowering the learning rate does not reduce the variance of the SGD updates, and thus does not reduce the noise term. This further indicates a regime of high gradient variance but low gradient confusion where SGD would converge faster than predicted by classical convergence rate results (Moulines & Bach, 2011; Needell et al., 2014).

In practice, typically the gradient variance is bounded. In this case, tighter convergence rate bounds in terms of $\eta$ are possible, which follow as simple corollaries from classical convergence bounds in terms of the gradient variance such as those in Moulines & Bach (2011); Needell et al. (2014). The *interpolation* and *strong growth* conditions, as analyzed in Ma et al. (2017); Bassily et al. (2018); Vaswani et al. (2018); Schmidt & Roux (2013), are related to this. These conditions imply that the variance of the gradients at the minimizer is small. As shown above, this would imply that the gradient confusion is also small, leading to fast convergence of SGD. This is also similar to the condition of the minimal risk being small as in Srebro et al. (2010); Zhang & Zhou (2019).

That being said, it is worth noting that the main contribution of this paper is to better understand how the network architecture impacts SGD. In general, it is not tractable to prove the concentration bounds in section 4 using the covariance matrix of the gradients alone without further unrealistic assumptions, such as infinitely wide networks (Schoenholz et al., 2016). A key contribution of this paper is to identify a suitable surrogate (i.e., the gradient confusion bound) to help study this relationship using new tight and clean bounds.

**Connections to gradient diversity:** In Yin et al. (2017), the authors define a property called *gradient diversity*. This quantity is related to gradient confusion, but with important differences. Gradient diversity also measures the degree to which individual gradients at different data samples are different from each other. However, the gradient diversity measure gets larger as the individual gradients become orthogonal to each other, and further increases as the gradients start pointing in opposite directions. In a large batch, higher gradient diversity is desirable, and this leads to improved convergence rates in distributed settings, as shown in Yin et al. (2017). On the other hand, gradient confusion between two individual gradients is zero unless the inner product between them is negative. This makes gradient confusion useful for studying convergence of small minibatch SGD. This is because different possible SGD updates do not conflict with each other unless they are negatively correlated with each other.

The choice of the definition of gradient diversity in Yin et al. (2017) has important implications when its behavior is studied in overparameterized settings. Chen et al. (2018) extends the work of Yin et al. (2017), where the authors prove on 2-layer neural nets (and multi-layer linear neural nets) that gradient diversity *increases* with increased width and decreased depth. This metric does not

however distinguish between the cases where gradients become more orthogonal vs. more negatively correlated. As we show in this paper, this can have very different effects on the convergence of SGD in overparameterized settings. Specifically, we show that increased width and decreased depth *decreases* gradient confusion, and makes these networks easier to train. In fact, we see that the gradients become more orthogonal to each other in this case as shown in section 5. Thus, we view our papers to be complementary to each other, providing insights about different issues (large batch distributed training vs. small minibatch convergence).

**Other work on the impact of neural net structure:** Other works have also studied the impact of structured gradients on SGD. Balduzzi et al. (2017) study the effects of *shattered gradients* at initialization for ReLU networks, which is when stochastic gradients at initialization become negatively correlated. The authors show how gradients get increasingly shattered with depth in ReLU networks. Hanin (2018) shows that the variance of gradients in fully connected networks with ReLU activations is exponential in the sum of the reciprocals of the hidden layer widths at initialization. Further, Hanin & Rolnick (2018) show that this sum of the reciprocals of the hidden layer widths determines the variance of the sizes of the activations at each layer during initialization. When this sum of reciprocals is too large, early training dynamics are very slow, suggesting the difficulties of starting training on deeper networks, as well as the benefits of increased width.

**Other work on SGD convergence on overparameterized neural nets:** The convergence of SGD on over-parameterized models has received a lot of attention recently. Arora et al. (2018b) study the behavior of SGD on over-parameterized problems, and show that SGD on over-parameterized linear neural nets is similar to applying a certain preconditioner while optimizing. This can sometimes lead to acceleration when overparameterizing by increasing the depth of linear neural networks. In this paper, we show that this property does not hold in general (as mentioned briefly in Arora et al. (2018b)), and that convergence typically slows down because of gradient confusion when training very deep networks. There has recently also been interest in analyzing conditions under which SGD converges to global minimizers of overparameterized linear and non-linear neural networks. Arora et al. (2018a) shows SGD converges linearly to global minimizers for linear neural nets under certain conditions. Du et al. (2018); Allen-Zhu et al. (2018); Zou et al. (2018); Brutzkus et al. (2017) also show convergence to global minimizers of SGD for non-linear nets. While all these results require the network to be sufficiently wide, they represent an important step in the direction of better understanding optimization on neural nets. This paper complements these recent results by studying how low gradient confusion contributes to SGD's success on modern overparameterized neural nets.

## 7 CONCLUSIONS

In this paper, we investigate how overparameterization and model architecture affect the dynamics of SGD on neural networks. To help formally analyze this, we introduce a concept called gradient confusion, and show that when gradient confusion is low, SGD experiences fast convergence. We then show that increasing layer width leads to lower gradient confusion, making the model easier to train. In contrast, increasing network depth results in higher gradient confusion, making deeper models harder to train. We further show how techniques like batch normalization and skip connections help in tackling this problem.

Our results provide a number of important insights that can be used for better neural net model design, as well as for better algorithms for better training and generalization. Our results on the test set accuracies in appendix A suggest that an interesting topic for future work would be to investigate the connection between gradient confusion and generalization (Fort et al., 2019). The difference in the gradient confusion within the same class and across classes could also be an interesting tool to study adversarial training. Note that many previous results have shown how deeper models are more efficient at modeling higher complexity function classes than wider models, and thus depth is essential for the success of neural networks (Eldan & Shamir, 2016; Telgarsky, 2016; Raghu et al., 2017). Our results indicate that, given a sufficiently deep network, increasing the network width is important for the trainability of the model, and will lead to faster convergence rates. This is further supported by other recent research (Hanin, 2018; Hanin & Rolnick, 2018) that suggest that the width should increase linearly with depth in a neural network to help dynamics at the beginning of training. Our results also suggest the importance of further investigation into good initialization schemes for neural networks that make training very deep models possible (Zhang et al., 2019).

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

# A   ADDITIONAL EXPERIMENTAL RESULTS

In this section, we present more details about our experimental setup, as well as, additional experimental results on a range of models (MLPs, CNNs and Wide ResNets) and a range of datasets (MNIST, CIFAR-10, CIFAR-100).

## A.1   MLPs ON MNIST

To further test the main claims in the paper, we performed additional experiments on an image classification problem on the MNIST dataset using fully connected neural networks. We iterated over neural networks of varying depth and width, and considered both the identity activation function (i.e., linear neural networks) and the tanh activation function. We also considered two different weight initializations that are popularly used and appropriate for these activation functions:

- The Glorot normal initializer (Glorot & Bengio, 2010) with weights initialized by sampling from the distribution $\mathcal{N}\big(0, 2/(\text{fan-in} + \text{fan-out})\big)$, where fan-in denotes the number of input units in the weight matrix, and fan-out denotes the number of output units in the weight matrix.

- The LeCun normal initializer (LeCun et al., 2012) with weights initialized by sampling from the distribution $\mathcal{N}\big(0, 1/\text{fan-in}\big)$.

We considered the simplified case where all hidden layers have the same width $\ell$. Thus, the first weight matrix $\mathbf{W}_0 \in \mathbb{R}^{\ell \times d}$, where $d = 784$ for the $28 \times 28$-sized images of MNIST; all intermediate weight matrices $\{\mathbf{W}_p\}_{p \in [\beta-1]} \in \mathbb{R}^{\ell \times \ell}$; and the final layer $\mathbf{W}_\beta \in \mathbb{R}^{10 \times \ell}$ for the 10 image classes in MNIST. We added biases to each layer, which we initialized to 0. We used softmax cross entropy as the loss function. We use MLP-$\beta$-$\ell$ to denote this fully connected network of depth $\beta$ and width $\ell$. We used the standard train-valid-test splits of 40000-10000-10000 for MNIST.

This relatively simple model gave us the ability to iterate over a large number of combinations of network architectures of varying width and depth, and different activation functions and weight initializations. Linear neural networks are an efficient way to directly understand the effect of changing depth and width without increasing model complexity over linear regression. Thus, we considered both linear and non-linear neural nets in our experiments.

We used SGD with constant learning rates for training with a mini-batch size of 128 and trained each model for 40000 iterations (more than 100 epochs). The constant learning rate $\alpha$ was tuned over a logarithmically-spaced grid:

$$\alpha \in \{10^0, 10^{-1}, 10^{-2}, 10^{-3}, 10^{-4}, 10^{-5}, 10^{-6}\}.$$

We ran each experiment 10 times (making sure at least 8 of them ran till completion), and picked the learning rate that achieved the lowest training loss value on average at the end of training. Our grid search was such that the optimal learning rate never occurred at one of the extreme values tested.

To measure gradient confusion at the end training, we sampled 1000 pairs of mini-batches each of size 128 (the same size as the training batch size). We calculated gradients on each of these pairs of mini-batches, and then calculated the cosine similarity between them. To measure the worse-case gradient confusion, we computed the lowest gradient cosine similarity among all pairs. We explored the effect of changing depth and changing width on the different activation functions and weight initializations. We plot the final training loss achieved for each model and the minimum gradient cosine similarities calculated over the 1000 pairs of gradients at the end of training. For each point, we plot both the mean and the standard deviation over the 10 independent runs.

**The effect of depth.** We first present results showing the effect of network depth. We considered a fixed width of $\ell = 100$, and varied the depth of the neural network, on the log scale, as:

$$\beta \in \{3, 10, 30, 100, 300, 1000\}.$$

Figure 5 shows results on neural networks with identity and tanh activation functions for the two weight initializations considered (Glorot normal and LeCun normal). Similar to the experimental results in section 5, and matching our theoretical results in section 4, we notice the consistent trend of gradient confusion increasing with increasing depth. This makes the networks harder to train with increasing depth, and this is evidenced by an increase in the final training loss value. By depth

$\beta = 1000$, the increased gradient confusion effectively makes the network untrainable when using tanh non-linearities.

**The effect of width.** We explored the effect of width by varying the width of the neural network while keeping the depth fixed at $\beta = 300$. We chose a very deep model, which is essentially untrainable for small widths (with standard initialization techniques) and helps better illustrate the effects of increasing width. We varied the width of the network, again on the log scale, as:

$$\ell \in \{10, 30, 100, 300, 1000\}.$$

Crucially, note that the smallest network considered here, MLP-300-10, still has more than 50000 parameters (i.e., more than the number of training samples), and the network with width $\ell = 30$ has almost three times the number of parameters as the high-performing MLP-3-100 network considered in the previous section. Figure 6 show results on linear neural nets and neural nets with tanh activations for both the Glorot normal and LeCun normal initializations. As in the experimental results of section 5, we see the consistent trend of gradient confusion decreasing with increasing width. Thus, wider networks become easier to train and improve the final training loss value. We further see that when the width is too small ($\ell = 30$), the gradient confusion becomes drastically high and the network becomes completely untrainable.

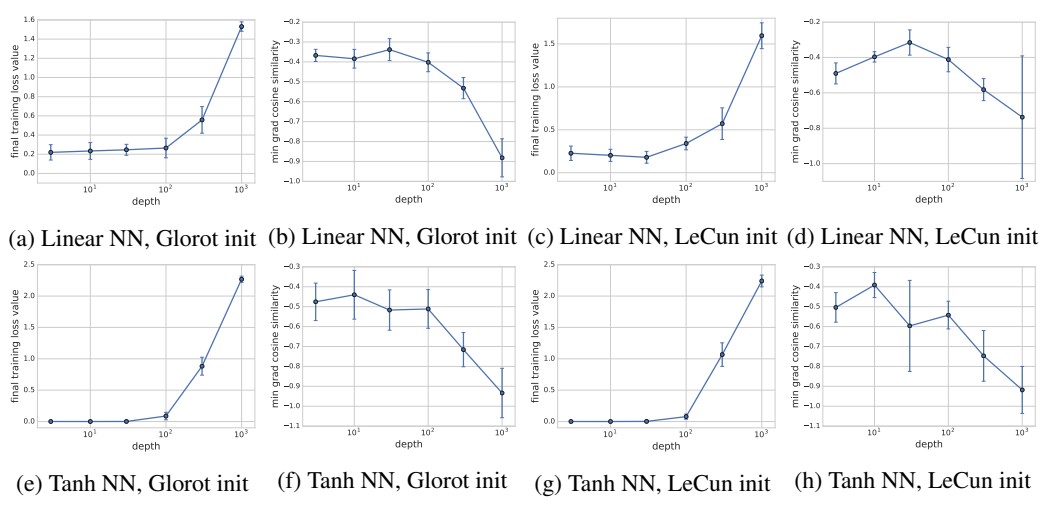

(a) Linear NN, Glorot init  (b) Linear NN, Glorot init  (c) Linear NN, LeCun init  (d) Linear NN, LeCun init

(e) Tanh NN, Glorot init  (f) Tanh NN, Glorot init  (g) Tanh NN, LeCun init  (h) Tanh NN, LeCun init

Figure 5: Effect of varying depth on MLP-$\beta$-100.

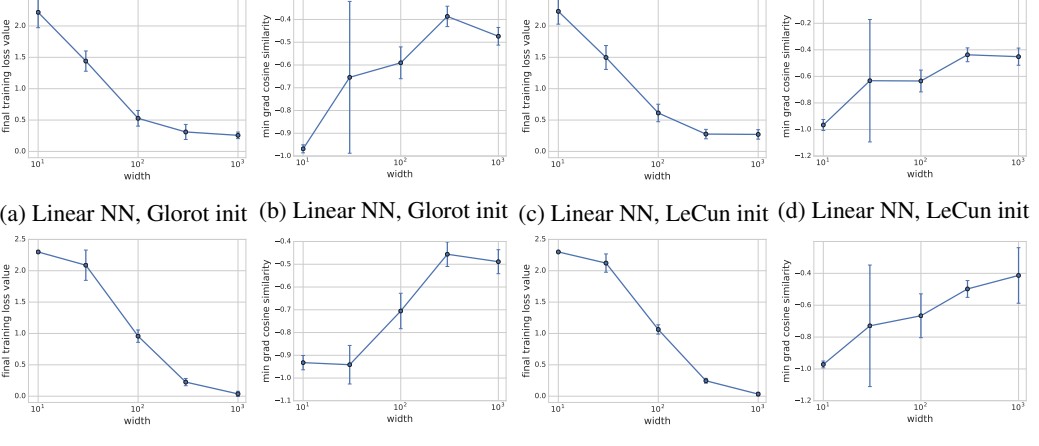

(a) Linear NN, Glorot init  (b) Linear NN, Glorot init  (c) Linear NN, LeCun init  (d) Linear NN, LeCun init

(e) Tanh NN, Glorot init  (f) Tanh NN, Glorot init  (g) Tanh NN, LeCun init  (h) Tanh NN, LeCun init

Figure 6: Effect of varying width on MLP-300-$\ell$.

## A.2 ADDITIONAL EXPERIMENTAL DETAILS FOR CNNS AND WRNS

In this section, we review the details of our setup for the image classification experiments on CNNs and WRNs on the CIFAR-10 and CIFAR-100 datasets.

### WIDE RESIDUAL NETWORKS

The Wide ResNet (WRN) architecture (Zagoruyko & Komodakis, 2016) for CIFAR datasets is a stack of three groups of residual blocks. There is a downsampling layer between two blocks, and the number of channels (width of a convolutional layer) is doubled after downsampling. In the three groups, the width of convolutional layers is $\{16\ell, 32\ell, 64\ell\}$, respectively. Each group contains $\beta_r$ residual blocks, and each residual block contains two $3 \times 3$ convolutional layers equipped with ReLU activation, batch normalization and dropout. There is a $3 \times 3$ convolutional layer with 16 channels before the three groups of residual blocks. And there is a global average pooling, a fully-connected layer and a softmax layer after the three groups. The depth of WRN is $\beta = 6\beta_r + 4$.

For our experiments, we turned off dropout. Unless otherwise specified, we also turned off batch normalization. We added biases to the convolutional layers when not using batch normalization to maintain model expressivity. We used the MSRA initializer (He et al., 2015) for the weights as is standard for this model, and used the same preprocessing steps for the CIFAR images as described in Zagoruyko & Komodakis (2016). This preprocessing step involves normalizing the images and doing data augmentation (Zagoruyko & Komodakis, 2016). We denote this network as WRN-$\beta$-$\ell$, where $\beta$ represents the depth and $\ell$ represents the width factor of the network.

To study the effect of depth, we considered WRNs with width factor $\ell = 2$ and depth varying as:

$$\beta \in \{16, 22, 28, 34, 40, 52, 76, 100\}.$$

For cleaner figures, we sometimes plot a subset of these results: $\beta \in \{16, 28, 40, 52, 76, 100\}$. To study the effect of width, we considered WRNs with depth $\beta = 16$ and width factor varying as:

$$\ell \in \{2, 3, 4, 5, 6\}.$$

### CONVOLUTIONAL NEURAL NETS

The WRN architecture contains skip connections that, as we show, help in training deep networks. To consider VGG-like convolutional networks, we consider a family of networks where we remove the skip connections from WRNs. Following the WRN convention, we denote these networks as CNN-$\beta$-$\ell$, where $\beta$ denotes the depth and $\ell$ denotes the width factor.

To study the effect of depth, we considered CNNs with width factor $\ell = 2$ and depth varying as:

$$\beta \in \{16, 22, 28, 34, 40\}.$$

To study the effect of width, we considered CNNs with depth $\beta = 16$ and width factor varying as:

$$\ell \in \{2, 3, 4, 5, 6\}.$$

### HYPERPARAMETER TUNING AND OTHER DETAILS

We used SGD as the optimizer without any momentum. Following Zagoruyko & Komodakis (2016), we ran all experiments for 200 epochs with minibatches of size 128, and reduced the initial learning rate by a factor of 10 at epochs 80 and 160. We turned off weight decay for all our experiments.

We ran each individual experiment 5 times. We ignored any runs that were unable to decrease the loss from its initial value. We also made sure at least 4 out of the 5 independent runs ran till completion. When the learning rate is close to the threshold at which training is still possible, some runs may converge, while others may fail to converge. Thus, these checks ensure that we pick a learning rate that converges reliably in most cases on each problem. We show the standard deviation across runs in our plots.

We tuned the optimal initial learning rate for each model over a logarithmically-spaced grid:

$$\alpha \in \{10^1, 3 \times 10^0, 10^0, 3 \times 10^{-1}, 10^{-1}, 3 \times 10^{-2}, 10^{-2}, 3 \times 10^{-3}, 10^{-3}, 3 \times 10^{-4}, 10^{-4}, 3 \times 10^{-5}\},$$

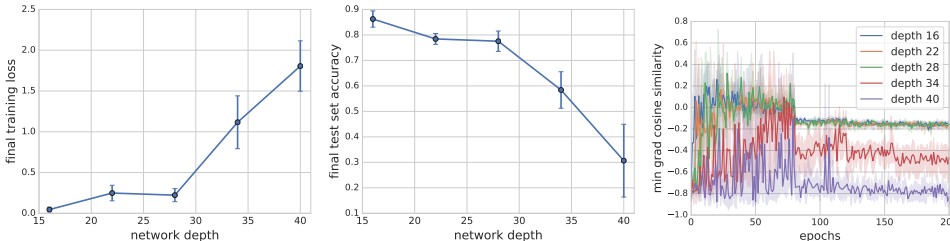

Figure 7: The effect of network depth with CNN-$\beta$-2 on CIFAR-10. *Left plot*: final training loss values at the end of training, *Middle plot*: final test set accuracy values at the end of training. *Right plot*: curves showing the minimum of pairwise gradient cosine similarities during training.

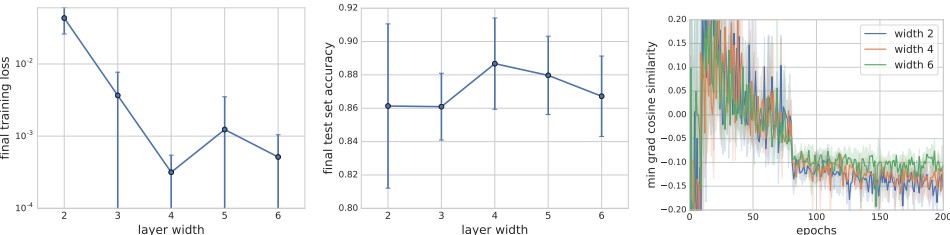

Figure 8: The effect of width with CNN-16-$\ell$ on CIFAR-10. *Left plot*: final training loss values at the end of training, *Middle plot*: final test set accuracy values at the end of training. *Right plot*: curves showing the minimum of pairwise gradient cosine similarities during training.

and selected the run that achieved the lowest final training loss value (averaged over the independent runs). Our grid search was such that the optimal learning rate never occurred at one of the extreme values tested. We used the standard train-valid-test splits of 40000-10000-10000 for CIFAR-10 and CIFAR-100.

To measure gradient confusion, at the end of every training epoch, we sampled 100 pairs of mini-batches each of size 128 (the same size as the training batch size). We calculated gradients on each mini-batch, and then computed pairwise cosine similarities. To measure the worse-case gradient confusion, we computed the lowest gradient cosine similarity among all pairs. We also show the kernel density estimation of the pairwise gradient cosine similarities of the 100 minibatches sampled at the end of training (after 200 epochs), to see the concentration of the distribution. To do this, we combine together the 100 samples for each independent run and then perform kernel density estimation with a gaussian kernel on this data.

### A.3  ADDITIONAL PLOTS FOR CIFAR-10 ON CNNS

In section 5, we showed results for image classification using CNNs on CIFAR-10. In this section, we show some additional plots for this experiment. Figure 7 shows the effect of changing the depth, while figure 8 shows the effect of changing the width factor of the CNN. We see that the final training loss and test set accuracy values show the same trends as in section 5: deeper networks are harder to train, while wider networks are easier to train. As mentioned previously, theorems 3.1 and 3.2 indicate that we would expect the effect of gradient confusion to be more prominent near the end of training. From the plots we see that deeper networks have higher gradient confusion close to minimum, while wider networks have lower gradient confusion close to the minimum.

### A.4  CIFAR-100 ON CNNS

We now consider image classifications tasks with CNNs on the CIFAR-100 dataset. Figure 9 shows the effect of varying depth, while figure 10 shows the effect of varying width. We notice the same trends as in our results with CNNs on CIFAR-10. Interestingly, from the width results in figure 10, we see that while there is no perceptible change to the minimum pairwise gradient cosine similarity, the distribution still sharply concentrates around 0 with increasing width. Thus more gradients become orthogonal to each other with increasing width, implying that SGD on very wide networks becomes closer to decoupling over the data samples.

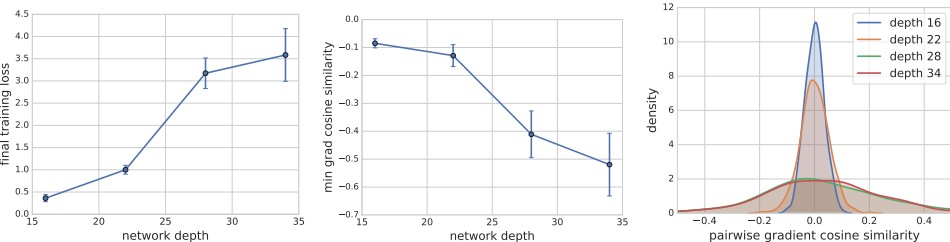

Figure 9: The effect of network depth with CNN-$\beta$-2 on CIFAR-100. *Left plot*: training loss values at the end of training. *Middle plot*: minimum of pairwise gradient cosine similarities at the end of training, *Right plot*: kernel density estimate of the pairwise gradient cosine similarities at the end of training.

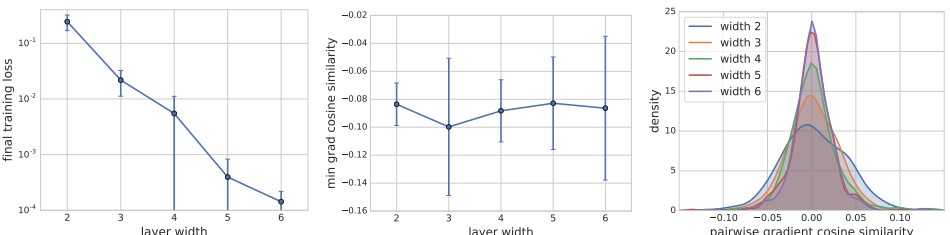

Figure 10: The effect of width with CNN-16-$\ell$ on CIFAR-100. *Left plot*: training loss values at the end of training. *Middle plot*: minimum of pairwise gradient cosine similarities at the end of training, *Right plot*: kernel density estimate of the pairwise gradient cosine similarities at the end of training.

### A.5 IMAGE CLASSIFICATION WITH WRNS ON CIFAR-10 AND CIFAR-100

We now show results for image classification problems using wide residual networks (WRNs) on CIFAR-10 and CIFAR-100. The WRNs we consider do not have any batch normalization. Later we show results on the effect of adding batch normalization to these networks.

Figures 11 and 12 show results on the effect of depth using WRNs on CIFAR-10 and CIFAR-100 respectively. We again see the consistent trend of deeper networks having higher gradient confusion, making them harder to train. We further see that increasing depth results in the pairwise gradient cosine similarities concentrating less around 0.

Figures 13 and 14 show results on the effect of width using WRNs on CIFAR-10 and CIFAR-100 respectively. We see that increasing width typically lowers gradient confusion and helps the network achieve lower loss values. The pairwise gradient cosine similarities also typically concentrate around 0 with higher width. We also notice from these figures that in some cases, increasing width might lead to diminishing returns, i.e., the benefits of increased width diminish after a certain point, as one would expect.

### A.6 EFFECT OF BATCH NORMALIZATION

In section 5 we showed results on the effect of adding batch normalization to CNNs and WRNs on an image classification task on CIFAR-10. In this section, we present similar results for image classification on CIFAR-100. Similar to section 5, figure 15 shows that adding skip connections or batch normalization individually help in training deeper models, but these models still suffer from worsening results and increasing gradient confusion as the network gets deeper. Both these techniques together keep the gradient confusion relatively low even for very deep networks, significantly improving trainability of deep models.

## B NEAR ORTHOGONALITY OF RANDOM VECTORS

For completeness, we state and prove below a lemma on the near orthogonality of random vectors. This result is often attributed to Milman & Schechtman (1986).

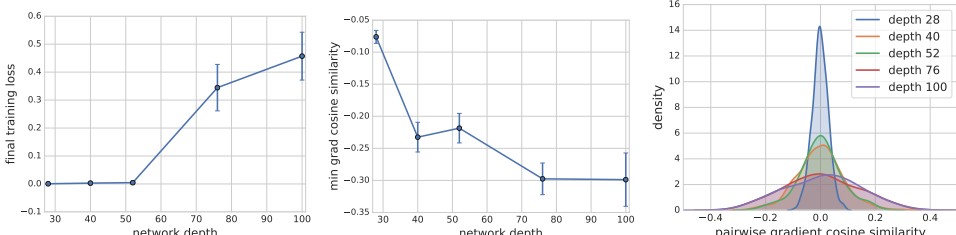

Figure 11: The effect of depth with WRN-$\beta$-2 (no batch normalization) on CIFAR-10. *Left plot*: training loss values at the end of training. *Middle plot*: minimum of pairwise gradient cosine similarities at the end of training, *Right plot*: kernel density estimate of the pairwise gradient cosine similarities at the end of training.

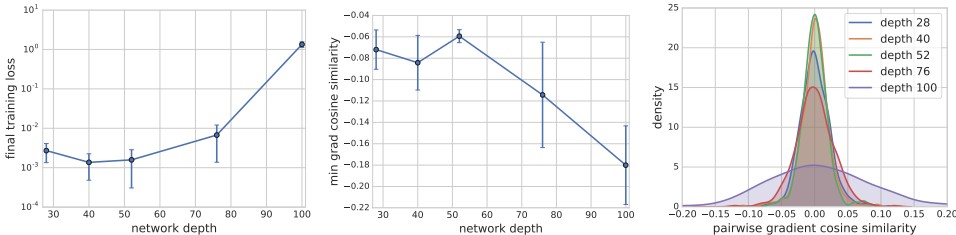

Figure 12: The effect of depth with WRN-$\beta$-2 (no batch normalization) on CIFAR-100. *Left plot*: training loss values at the end of training. *Middle plot*: minimum of pairwise gradient cosine similarities at the end of training, *Right plot*: kernel density estimate of the pairwise gradient cosine similarities at the end of training.

**Lemma B.1** (Near orthogonality of random vectors). *For vectors $\{\mathbf{x_i}\}_{i \in [N]}$ drawn uniformly from a unit sphere in $d$ dimensions, and $\nu > 0$,*

$$\Pr\left[\exists i, j \ |\mathbf{x}_i^\top \mathbf{x}_j| > \nu\right] \leq N^2 \sqrt{\tfrac{\pi}{8}} \exp\left(-\tfrac{d-1}{2}\nu^2\right).$$

*Proof.* Given a fixed vector $\mathbf{x}$, a uniform random vector $\mathbf{y}$ satisfies $|\mathbf{x}^\top \mathbf{y}| \geq \nu$ only if $\mathbf{y}$ lies in one of two spherical caps: one centered at $\mathbf{x}$ and the other at $-\mathbf{x}$, and both with angular radius $\cos^{-1}(\nu) \leq \frac{\pi}{2} - \nu$. A simple result often attributed to Milman & Schechtman (1986) bounds the probability of lying in either of these caps as

$$\Pr[|\mathbf{x}^\top \mathbf{y}| \geq \nu] \leq \sqrt{\frac{\pi}{2}} \exp\left(-\frac{d-1}{2}\nu^2\right). \tag{5}$$

Because of rotational symmetry, the bound (5) holds if both $\mathbf{x}$ and $\mathbf{y}$ are chosen uniformly at random.

We next apply a union bound to control the probability that $|\mathbf{x}_i^\top \mathbf{x}_j| \geq \nu$ for some pair $(i, j)$. There are fewer than $N^2/2$ such pairs, and so the probability of this condition is

$$\Pr[|\mathbf{x}_i^\top \mathbf{x}_j| \geq \nu, \text{for some } i, j] \leq \frac{N^2}{2} \sqrt{\frac{\pi}{2}} \exp\left(-\frac{d-1}{2}\nu^2\right). \qquad \square$$

## C  LOW-RANK HESSIANS LEAD TO LOW GRADIENT CONFUSION

In this section, we slightly elaborate on the argument that low-rank random Hessians result in low gradient confusion. For clarity in presentation, suppose each $f_i$ has a minimizer at the origin (the same argument can be easily extended to the more general case). Suppose also that there is a Lipschitz constant for the Hessian of each function $f_i$ that satisfies $\|\mathbf{H}_i(\mathbf{w}) - \mathbf{H}_i(\mathbf{w}')\| \leq L_H \|\mathbf{w} - \mathbf{w}'\|$. Then $\nabla f_i(\mathbf{w}) = \mathbf{H}_i\mathbf{w} + \mathbf{e}$, where $\mathbf{e}$ is an error term bounded as: $\|\mathbf{e}\| \leq \frac{1}{2}L_H\|\mathbf{w}\|^2$, and we use the shorthand $\mathbf{H}_i$ to denote $\mathbf{H}_i(\mathbf{0})$. Then we have:

$$\begin{aligned}
|\langle \nabla f_i(\mathbf{w}), \nabla f_j(\mathbf{w}) \rangle| &= |\langle \mathbf{H}_i\mathbf{w}, \mathbf{H}_j\mathbf{w} \rangle| + \langle \mathbf{e}, \mathbf{H}_i\mathbf{w} + \mathbf{H}_j\mathbf{w} \rangle + \|\mathbf{e}\|^2 \\
&\leq \|\mathbf{w}\|^2 \|\mathbf{H}_i\| \|\mathbf{H}_j\| + \|\mathbf{e}\| \|\mathbf{w}\| (\|\mathbf{H}_i\| + \|\mathbf{H}_j\|) + \|\mathbf{e}\|^2 \\
&\leq \|\mathbf{w}\|^2 \|\mathbf{H}_i\| \|\mathbf{H}_j\| + \frac{1}{2} L_H \|\mathbf{w}\|^3 (\|\mathbf{H}_i\| + \|\mathbf{H}_j\|) + \frac{1}{4} L_H^2 \|\mathbf{w}\|^4.
\end{aligned}$$

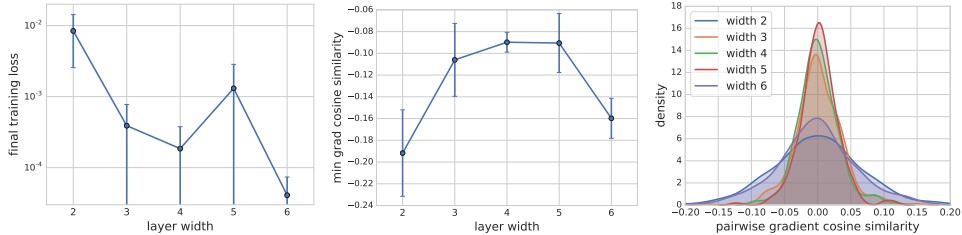

Figure 13: The effect of width with WRN-16-$\ell$ (no batch normalization) on CIFAR-10. *Left plot*: training loss values at the end of training. *Middle plot*: minimum of pairwise gradient cosine similarities at the end of training, *Right plot*: kernel density estimate of the pairwise gradient cosine similarities at the end of training.

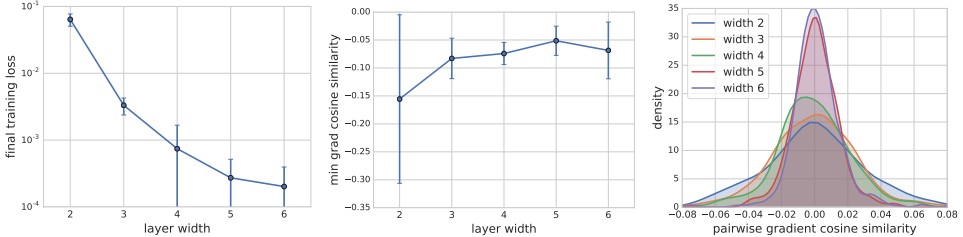

Figure 14: The effect of width with WRN-16-$\ell$ (no batch normalization) on CIFAR-100. *Left plot*: training loss values at the end of training. *Middle plot*: minimum of pairwise gradient cosine similarities at the end of training, *Right plot*: kernel density estimate of the pairwise gradient cosine similarities at the end of training.

If the Hessians are sufficiently random and low-rank (e.g., of the form $\mathbf{H}_i = \mathbf{a}_i \mathbf{a}_i^\top$ where $\mathbf{a}_i \in \mathbb{R}^{N \times r}$ are randomly sampled from a unit sphere), then one would expect the terms in this expression to be small for all $\mathbf{w}$ within a neighborhood of the minimizer. Given that it has been observed that many standard neural nets have low rank Hessians at the minimizer, this indicates that the gradient confusion might be low for a large class of weights near the minimizer for these models.

# D  Missing proofs

## D.1  Proofs of theorems 3.1 and 3.2

This section presents proofs for the convergence theorems of SGD presented in section 3, under the assumption of low gradient confusion. For clarity of presentation, we re-state each theorem before its proof.

**Theorem 3.1.** *If the objective function satisfies (A1) and (A2), and has gradient confusion $\eta$, SGD with updates of the form* (2) *converges linearly to a neighborhood of the minima of problem* (1) *as:*

$$\mathbb{E}[F(\mathbf{w}_T) - F^\star] \leq \rho^T (F(\mathbf{w}_0) - F^\star) + \frac{\alpha \eta}{1 - \rho},$$

*where $\alpha < \frac{2}{NL}$, $\rho = 1 - \frac{2\mu}{N}\left(\alpha - \frac{NL\alpha^2}{2}\right)$, $F^\star = \min_{\mathbf{w}} F(\mathbf{w})$ and $\mathbf{w}_0$ is the initialized weights.*

*Proof.* Let $\tilde{i} \in [N]$ denote the index of the realized function $\tilde{f}_k$ in the uniform sampling from $\{f_i\}_{i \in [N]}$ at step $k$. From assumption (A1), we have

$$F(\mathbf{w}_{k+1}) \leq F(\mathbf{w}_k) + \langle \nabla F(\mathbf{w}_k), \mathbf{w}_{k+1} - \mathbf{w}_k \rangle + \frac{L}{2} \|\mathbf{w}_{k+1} - \mathbf{w}_k\|^2$$

$$= F(\mathbf{w}_k) - \alpha \langle \nabla F(\mathbf{w}_k), \nabla \tilde{f}_k(\mathbf{w}_k) \rangle + \frac{L\alpha^2}{2} \|\nabla \tilde{f}_k(\mathbf{w}_k)\|^2$$

$$= F(\mathbf{w}_k) - \left(\frac{\alpha}{N} - \frac{L\alpha^2}{2}\right) \|\nabla \tilde{f}_k(\mathbf{w}_k)\|^2 - \frac{\alpha}{N} \sum_{\forall i : i \neq \tilde{i}} \langle \nabla f_i(\mathbf{w}_k), \nabla \tilde{f}_k(\mathbf{w}_k) \rangle$$

$$\leq F(\mathbf{w}_k) - \left(\frac{\alpha}{N} - \frac{L\alpha^2}{2}\right) \|\nabla \tilde{f}_k(\mathbf{w}_k)\|^2 + \frac{\alpha(N-1)\eta}{N},$$

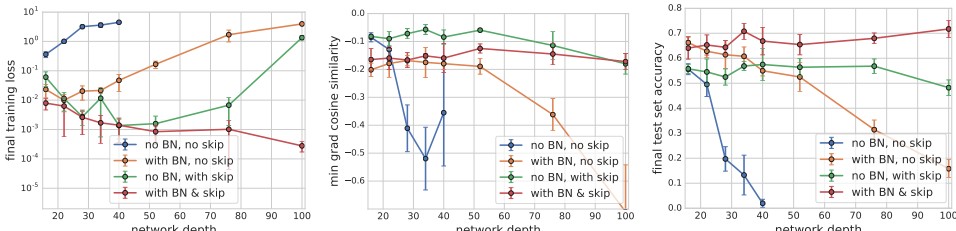

Figure 15: The effect of adding skip connections and batch normalization to CNN-$\beta$-2 on CIFAR-100. Plots show the training loss, minimum pairwise gradient cosine similarities, and test accuracies at the end of training.

$$\leq F(\mathbf{w}_k) - \Big(\frac{\alpha}{N} - \frac{L\alpha^2}{2}\Big)\|\nabla \tilde{f}_k(\mathbf{w}_k)\|^2 + \alpha\eta,$$

where the second-last inequality follows from definition 2.1. Let the learning rate $\alpha < 2/NL$. Then, using assumption (A2) and subtracting by $F^\star = \min_{\mathbf{w}} F(\mathbf{w})$ on both sides, we get

$$F(\mathbf{w}_{k+1}) - F^\star \leq F(\mathbf{w}_k) - F^\star - 2\mu\Big(\frac{\alpha}{N} - \frac{L\alpha^2}{2}\Big)(\tilde{f}_k(\mathbf{w}_k) - \tilde{f}_k^\star) + \alpha\eta,$$

where $\tilde{f}_k^\star = \min_{\mathbf{w}} \tilde{f}_k(\mathbf{w})$. It is easy to see that by definition we have, $\mathbb{E}_i[f_i^\star] \leq F^\star$. Moreover, from assumption that $\alpha < \frac{2}{NL}$, it implies that $\Big(\frac{\alpha}{N} - \frac{L\alpha^2}{2}\Big) > 0$. Therefore, taking expectation on both sides we get,

$$\mathbb{E}[F(\mathbf{w}_{k+1}) - F^\star] \leq \Big(1 - \frac{2\mu\alpha}{N} + \mu L\alpha^2\Big)\mathbb{E}[F(\mathbf{w}_k) - F^\star] + \alpha\eta.$$

Writing $\rho = 1 - \frac{2\mu\alpha}{N} + \mu L\alpha^2$, and unrolling the iterations, we get

$$\mathbb{E}[F(\mathbf{w}_{k+1}) - F^\star] \leq \rho^{k+1}(F(\mathbf{w}_0) - F^\star) + \sum_{i=0}^{k} \rho^i \alpha\eta$$

$$\leq \rho^{k+1}(F(\mathbf{w}_0) - F^\star) + \sum_{i=0}^{\infty} \rho^i \alpha\eta$$

$$= \rho^{k+1}(F(\mathbf{w}_0) - F^\star) + \frac{\alpha\eta}{1-\rho}. \qquad \square$$

**Theorem 3.2.** *If the objective satisfies (A1) and has gradient confusion bound $\eta$, then SGD converges to a neighborhood of a stationary point as:*

$$\min_{k=1,\ldots,T} \mathbb{E}\|\nabla F(\mathbf{w}_k)\|^2 \leq \frac{\rho(F(\mathbf{w}_1) - F^\star)}{T} + \rho\eta,$$

*for learning rate $\alpha < \frac{2}{NL}$, $\rho = \frac{2N}{2-NL\alpha}$, and $F^\star = \min_{\mathbf{w}} F(\mathbf{w})$.*

*Proof.* From theorem 3.1, we have:

$$F(\mathbf{w}_{k+1}) \leq F(\mathbf{w}_k) - \Big(\frac{\alpha}{N} - \frac{L\alpha^2}{2}\Big)\|\nabla \tilde{f}_k(\mathbf{w}_k)\|^2 + \alpha\eta. \qquad (6)$$

Now we know that:

$$\mathbb{E}\|\nabla \tilde{f}_k(\mathbf{w}_k)\|^2 = \mathbb{E}\|\nabla \tilde{f}_k(\mathbf{w}_k) - \nabla F(\mathbf{w}_k)\|^2 + \mathbb{E}\|\nabla F(\mathbf{w}_k)\|^2 \geq \mathbb{E}\|\nabla F(\mathbf{w}_k)\|^2.$$

Thus, taking expectation and assuming the step size $\alpha < 2/(NL)$, we can rewrite equation 6 as:

$$\mathbb{E}\|\nabla F(\mathbf{w}_k)\|^2 \leq \frac{2N}{2\alpha - NL\alpha^2}\mathbb{E}[F(\mathbf{w}_k) - F(\mathbf{w}_{k+1})] + \frac{2N\eta}{2 - NL\alpha}.$$

Taking an average over $T$ iterations, and using $F^\star = \min_{\mathbf{w}} F(\mathbf{w})$, we get:

$$\min_{k=1,\ldots,T} \mathbb{E}\|\nabla F(\mathbf{w}_k)\|^2 \leq \frac{1}{T}\sum_{k=1}^{T} \mathbb{E}\|\nabla F(\mathbf{w}_k)\|^2 \leq \frac{2N}{2\alpha - NL\alpha^2}\frac{F(\mathbf{w}_1) - F^\star}{T} + \frac{2N\eta}{2 - NL\alpha}. \quad \square$$

## D.2 PROOFS OF HELPER LEMMAS

**Lemma D.1.** *Consider the set of loss-functions $\{f_i(\mathbf{W})\}_{i\in[N]}$ where all $f_i$ are either the square-loss function or the logistic-loss function. Consider a feed-forward neural network as defined in equation 4 whose weights $\mathbf{W}$ satisfy assumption 1,. Consider the gradient $\nabla_{\mathbf{W}} f_i(\mathbf{W})$ of each function $f_i$. Note that we can write $\nabla_{\mathbf{W}} f_i(\mathbf{W}) = \zeta_{\mathbf{x}_i}(\mathbf{W})\nabla_{\mathbf{W}} g_{\mathbf{W}}(\mathbf{x}_i)$, where we define $\zeta_{\mathbf{x}_i}(\mathbf{W}) = \partial f_i(\mathbf{W})/\partial g_{\mathbf{W}}$. Then we have the following properties.*

1. *When $\|\mathbf{x}\| \le 1$ we have $\|\nabla_{\mathbf{W}} g_{\mathbf{W}}(\mathbf{x}_i)\| \le 1$.*

2. *There exists a constant $\zeta_0 > 0$ such that $|\zeta_{\mathbf{x}_i}(\mathbf{W})| \le 2$, $\|\nabla_{\mathbf{x}_i}\zeta_{\mathbf{x}_i}(\mathbf{W})\|_2 \le \zeta_0$, $\|\nabla_{\mathbf{W}}\zeta_{\mathbf{x}_i}(\mathbf{W})\|_2 \le \zeta_0$.*

*We show that $\zeta_0 \le 2\sqrt{\beta}$, where $\beta$ is the depth of the neural network.*

*Proof.* Let $\mathbf{W}$ denote the tuple $(\mathbf{W}_p)_{p\in[\beta]_0}$. Consider $|\zeta_{\mathbf{x}_i}(\mathbf{W})| = |\partial f_i(\mathbf{W})/\partial g_{\mathbf{W}}|$. In the case of square-loss function this evaluates to $|g_{\mathbf{W}}(\mathbf{x}) - \mathcal{C}(\mathbf{x})| \le 2$. In case of logistic regression, this evaluates to $|\frac{-1}{1+\exp(\mathcal{C}(\mathbf{x}_i)g_{\mathbf{W}}(\mathbf{x}_i))}| \le 1$. Now we consider $\|\nabla_{\mathbf{x}_i}\zeta_{\mathbf{x}_i}(\mathbf{W})\|$. Consider the squared loss function. We then have the following.

$$\|\nabla_{\mathbf{x}_i}\zeta_{\mathbf{x}_i}(\mathbf{W})\| = \|\nabla_{\mathbf{x}_i} f'(\mathbf{W})\|$$
$$= \|\nabla_{\mathbf{x}_i} g_{\mathbf{W}}(\mathbf{x}_i) - \mathcal{C}(\mathbf{x}_i)\|$$
$$\le \|\nabla_{\mathbf{x}_i} g_{\mathbf{W}}(\mathbf{x}_i)\| + 1.$$

Likewise, consider the logistic-loss function. We then have the following.

$$\|\nabla_{\mathbf{x}_i}\zeta_{\mathbf{x}_i}(\mathbf{W})\| \le \left\| \frac{\mathcal{C}(\mathbf{x}_i)^2}{(1+\exp(\mathcal{C}(\mathbf{x}_i)g_{\mathbf{W}}(\mathbf{x}_i)))^2} \exp(\mathcal{C}(\mathbf{x}_i)g_{\mathbf{W}}(\mathbf{x}_i)) \right\| \|\nabla_{\mathbf{x}_i} g_{\mathbf{W}}(\mathbf{x}_i)\|$$
$$\le \|\nabla_{\mathbf{x}_i} g_{\mathbf{W}}(\mathbf{x}_i)\|.$$

Thus, it suffices to bound $\|\nabla_{\mathbf{x}_i} g_{\mathbf{W}}(\mathbf{x}_i)\|$. Using assumption 1 and the properties (P1), (P2) of $\sigma$, this can be upper-bounded by 1.

Consider $\nabla_{\mathbf{W}_p}\zeta_{\mathbf{x}_i}(\mathbf{W})$ for some layer index $p \in [\beta]_0$. We will show that $\|\nabla_{\mathbf{W}_p}\zeta_{\mathbf{x}_i}(\mathbf{W})\|_2 \le 2$. Then it immediately follows that $\|\nabla_{\mathbf{W}}\zeta_{\mathbf{x}_i}(\mathbf{W})\|_2 \le 2\sqrt{\beta}$. In the case of a squared loss function. We have the following.

$$\|\nabla_{\mathbf{W}_p}\zeta_{\mathbf{x}_i}(\mathbf{W})\| = \|\nabla_{\mathbf{W}_p} f'(\mathbf{W})\|$$
$$= \|\nabla_{\mathbf{W}_p} g_{\mathbf{W}}(\mathbf{x}_i) - \mathcal{C}(\mathbf{x}_i)\|$$
$$\le \|\nabla_{\mathbf{W}_p} g_{\mathbf{W}}(\mathbf{x}_i)\| + 1.$$

Likewise, consider the logistic-loss function. We then have the following.

$$\|\nabla_{\mathbf{W}_p}\zeta_{\mathbf{x}_i}(\mathbf{W})\| \le \left\| \frac{\mathcal{C}(\mathbf{x}_i)^2}{(1+\exp(\mathcal{C}(\mathbf{x}_i)g_{\mathbf{W}}(\mathbf{x}_i)))^2} \exp(\mathcal{C}(\mathbf{x}_i)g_{\mathbf{W}}(\mathbf{x}_i)) \right\| \|\nabla_{\mathbf{W}_p} g_{\mathbf{W}}(\mathbf{x}_i)\|$$
$$\le \|\nabla_{\mathbf{W}_p} g_{\mathbf{W}}(\mathbf{x}_i)\|.$$

Since $\|\nabla_{\mathbf{W}_p} g_{\mathbf{W}}(\mathbf{x}_i)\| \le 1$, we have that $\|\nabla_{\mathbf{W}_p}\zeta_{\mathbf{x}_i}(\mathbf{W})\| \le 2$ in both the cases. Thus, $\zeta_0 = 2\sqrt{\beta}$. $\qquad\square$

## D.3 PROOFS OF THEOREM 4.1 AND COROLLARY 4.1

In this section, we will analyze the case when we have non-linear activations at each layer.

**Theorem 4.1.** *Consider the problem of training neural nets (equation 4) using either the square-loss or the logistic-loss function. Let $\eta > 0$ be a given constant. Let the weights satisfy assumption 1 and the non-linearities in each layer satisfy properties (P1) and (P2). For some fixed constant $c > 0$, the gradient confusion bound in equation 3 holds with probability at least*

$$1 - N^2 \exp\left(\frac{-cd\eta^2}{16\zeta_0^4(\beta+2)^4}\right),$$

*For both the square-loss and the logistic-loss functions, $\zeta_0 \le 2\sqrt{\beta}$ (from lemma D.1).*

*Proof.* We show two key properties, namely bounded gradient and non negative expectation. We will then use both these properties to complete the proof.

**Bounded gradient.** For every $i \in [n]$ define $\zeta_{\mathbf{x}_i}(\mathbf{W}) := f'(\mathbf{W})$. For every $p \in [\beta]$ define $\mathbf{H}_p$ as follows.

$$\mathbf{H}_p(\mathbf{x}) := \sigma(\mathbf{W}_p \cdot \sigma(\mathbf{W}_{p-1} \cdot \sigma(\dots \cdot \sigma(\mathbf{W}_0 \cdot \mathbf{x})\dots).$$

Fix an $i \in [N]$. Then we have the following recurrence

$$g_\beta(\mathbf{x}_i) := \sigma'(H_\beta(\mathbf{x}_i))$$
$$\mathbf{g}_p(\mathbf{x}_i) := (\mathbf{W}_{p+1}^\top \cdot \mathbf{g}_{p+1}(\mathbf{x}_i)) \cdot \mathrm{Diag}(\sigma'(\mathbf{H}_p(\mathbf{x}_i))) \qquad \forall p \in \{0, 1, \dots, \beta-1\}.$$

Then the gradients can be written in terms of the above quantities as follows.

$$\nabla_{\mathbf{W}_p} f_i(\mathbf{W}) = \mathbf{g}_p(\mathbf{x}_i) \cdot \mathbf{H}_{p-1}(\mathbf{x}_i)^\top \qquad \forall p \in [\beta]_0.$$

We can write $h_{\mathbf{W}}(\mathbf{x}_i, \mathbf{x}_j)$ as follows.

$$\zeta_{\mathbf{x}_i}(\mathbf{W})\zeta_{\mathbf{x}_j}(\mathbf{W}) \left( \sum_{p \in [\beta]_0} \mathrm{Tr}[\mathbf{H}_{p-1}(\mathbf{x}_i) \cdot \mathbf{g}_p(\mathbf{x}_i)^\top \cdot \mathbf{g}_p(\mathbf{x}_j) \cdot \mathbf{H}_{p-1}(\mathbf{x}_i)^\top] \right). \tag{7}$$

We will now bound $\|\nabla_{(\mathbf{x}_i, \mathbf{x}_j)} h_{\mathbf{W}}(\mathbf{x}_i, \mathbf{x}_j)\|_2$. Consider $\nabla_{\mathbf{x}_i} h_{\mathbf{W}}(\mathbf{x}_i, \mathbf{x}_j)$. This can be written as follows.

$$(\nabla_{\mathbf{x}_i}\zeta_{\mathbf{x}_i}(\mathbf{W}))\zeta_{\mathbf{x}_j}(\mathbf{W}) \left( \sum_{p \in [\beta]_0} \mathrm{Tr}[\mathbf{H}_{p-1}(\mathbf{x}_i) \cdot \mathbf{g}_p(\mathbf{x}_i)^\top \cdot \mathbf{g}_p(\mathbf{x}_j) \cdot \mathbf{H}_{p-1}(\mathbf{x}_i)^\top] \right) +$$
$$\zeta_{\mathbf{x}_i}(\mathbf{W})\zeta_{\mathbf{x}_j}(\mathbf{W}) \sum_{p \in [\beta]_0} \left[ \nabla_{\mathbf{x}_i} \left( \mathbf{H}_{p-1}(\mathbf{x}_i) \cdot \mathbf{g}_p(\mathbf{x}_i)^\top \cdot \mathbf{g}_p(\mathbf{x}_j) \cdot \mathbf{H}_{p-1}(\mathbf{x}_i) \right) \right]^\top. \tag{8}$$

Observe that each of the entries in the diagonal matrix $\mathrm{Diag}(\sigma'(\mathbf{H}_p(\mathbf{x}_i)))$ is at most 1. Thus, we have that $\| \mathrm{Diag}(\sigma'(\mathbf{H}_p(\mathbf{x}_i)))\| \le 1$.

We have the following relationship.

$$\|g_\beta(\mathbf{x}_i)\| \le 1$$
$$\|\mathbf{g}_p(\mathbf{x}_i)\| \le \|\mathbf{W}_{p+1}^\top\|\|\mathbf{g}_{p+1}(\mathbf{x}_i))\|\| \mathrm{Diag}(\sigma'(\mathbf{H}_p(\mathbf{x}_i)))\| \le 1 \qquad \forall p \in \{0, 1, \dots, \beta-1\}.$$

Moreover we have,

$$\| \mathrm{Tr}[\mathbf{H}_{p-1}(\mathbf{x}_i) \cdot \mathbf{g}_p(\mathbf{x}_i)^\top \cdot \mathbf{g}_p(\mathbf{x}_j) \cdot \mathbf{H}_{p-1}(\mathbf{x}_i)^\top]\| \le \|\mathbf{H}_{p-1}(\mathbf{x}_i)\|\|\mathbf{g}_p(\mathbf{x}_i)^\top\|\|\mathbf{g}_p(\mathbf{x}_j)\|\|\mathbf{H}_{p-1}(\mathbf{x}_i)^\top\| \le 1.$$

Consider $\|\nabla_{\mathbf{x}_i} \left( \mathbf{H}_{p-1}(\mathbf{x}_i) \cdot \mathbf{g}_p(\mathbf{x}_i)^\top \cdot \mathbf{g}_p(\mathbf{x}_j) \cdot \mathbf{H}_{p-1}(\mathbf{x}_i) \right) \|$ for every $p \in [\beta]_0$.

This can be upper-bounded by,

$$\|\nabla_{\mathbf{x}_i}\mathbf{H}_{p-1}(\mathbf{x}_i)\|\|\mathbf{g}_p(\mathbf{x}_i)^\top\|\|\mathbf{g}_p(\mathbf{x}_j)\|\|\mathbf{H}_{p-1}(\mathbf{x}_i)\| + \|\mathbf{H}_{p-1}(\mathbf{x}_i)\|\|\nabla_{\mathbf{x}_i}\mathbf{g}_p(\mathbf{x}_i)^\top\|\|\mathbf{g}_p(\mathbf{x}_j)\|\|\mathbf{H}_{p-1}(\mathbf{x}_i)\|.$$

Note that $\nabla_{\mathbf{x}_i}\mathbf{H}_{p-1}(\mathbf{x}_i) = \mathbf{g}_1(\mathbf{x}_i) \cdot \mathrm{Diag}(\sigma'(\mathbf{W}_0 \cdot \mathbf{x}_i)) \cdot \mathbf{W}_0^\top \cdot \mathbf{g}_p(\mathbf{x}_i)^\top$. Thus, $\|\nabla_{\mathbf{x}_i}\mathbf{H}_{p-1}(\mathbf{x}_i)\| \le 1$. We will now show that $\|\nabla_{\mathbf{x}_i}\mathbf{g}_p(\mathbf{x}_i)\| \le \beta - p + 1$. We prove this inductively. Consider the base case when $p = \beta$.

$$\|\nabla_{\mathbf{x}_i}\mathbf{g}_\beta(\mathbf{x}_i)\| = \|\nabla_{\mathbf{x}_i}\sigma'(\mathbf{H}_\beta(\mathbf{x}_i))\| \le 1 = \beta - \beta + 1.$$

Now, the inductive step.

$$\|\nabla_{\mathbf{x}_i}\mathbf{g}_p(\mathbf{x}_i)\| \le \|\nabla_{\mathbf{x}_i}\mathbf{g}_{p+1}(\mathbf{x}_i)\| + \|\nabla_{\mathbf{x}_i}\mathrm{Diag}(\sigma'(\mathbf{H}_p(\mathbf{x}_i)))\| \le \beta - p \le \beta - p + 1.$$

Thus, using equation 8 and the above arguments, we obtain, $\|\nabla_{\mathbf{x}_i} h_{\mathbf{W}}(\mathbf{x}_i, \mathbf{x}_j)\|_2 \le \zeta_0^2(\beta+1) + \zeta_0^2(\beta+1)(\beta+2) \le 2\zeta_0^2(\beta+2)^2$ and thus, $\|\nabla_{(\mathbf{x}_i, \mathbf{x}_j)} h_{\mathbf{W}}(\mathbf{x}_i, \mathbf{x}_j)\|_2 \le 4\zeta_0^2(\beta+2)^2$.

**Non-negative expectation.**

$$\mathbb{E}_{\mathbf{x}_i,\mathbf{x}_j}[h(\mathbf{x}_i,\mathbf{x}_j)] = \mathbb{E}_{\mathbf{x}_i,\mathbf{x}_j}[\langle\nabla f_i(\mathbf{W}),\nabla f_j(\mathbf{W})\rangle]$$
$$= \langle\mathbb{E}_{\mathbf{x}_i}[\nabla f_i(\mathbf{W})],\mathbb{E}_{\mathbf{x}_j}[\nabla f_j(\mathbf{W})]\rangle$$
$$= \|\mathbb{E}_{\mathbf{x}_i}[\nabla f_i(\mathbf{W})]\|^2 \geq 0.$$

We have used the fact that $\nabla f_i(\mathbf{W})$ and $\nabla f_j(\mathbf{W})$ are identically distributed and independent.

**Concentration of Measure.** We combine the two properties as follows. From **Non-negative Expectation** property and equation 23, we have that

$$\Pr[h_\mathbf{W}(\mathbf{x}_i,\mathbf{x}_j) \leq -\eta] \leq \Pr[h_\mathbf{W}(\mathbf{x}_i,\mathbf{x}_j) \leq \mathbb{E}_{(\mathbf{x}_i,\mathbf{x}_j)}[h_\mathbf{W}(\mathbf{x}_i,\mathbf{x}_j)] - \eta] \leq \exp\left(\frac{-cd\eta^2}{16\zeta_0^4(\beta+2)^4}\right).$$
$$\tag{9}$$

To obtain the probability that *some* value of $h_\mathbf{w}(\nabla_\mathbf{w}f_i,\nabla_\mathbf{w}f_j)$ lies below $-\eta$, we use a union bound. There are $N(N-1)/2 < N^2/2$ possible pairs of data points to consider, and so this probability is bounded above by $N^2\exp\left(\frac{-cd\eta^2}{16\zeta_0^4(\beta+2)^4}\right)$. $\qquad\square$

### D.3.1 PROOF OF COROLLARY 4.1

Before we prove corollary 4.1 we prove the following helper lemma.

**Lemma D.2.** *Suppose* $\max_\mathbf{W}\|\nabla_\mathbf{W}f_i(\mathbf{W})\| \leq M$, *and both* $\nabla_\mathbf{W}f_i(\mathbf{w})$ *and* $\nabla_\mathbf{W}f_j(\mathbf{W})$ *are Lipschitz in* $\mathbf{W}$ *with constant* $L$. *Then* $h_\mathbf{W}(\mathbf{x}_i,\mathbf{x}_j)$ *is Lipschitz in* $\mathbf{W}$ *with constant* $2LM$.

*Proof.* We view $\mathbf{W}$ as flattened vector. We now prove the above result for these two vectors. For two vectors $\mathbf{w},\mathbf{w}'$,

$$|h_\mathbf{w}(\mathbf{x}_i,\mathbf{x}_j) - h_{\mathbf{w}'}(\mathbf{x}_i,\mathbf{x}_j)|$$
$$= |\langle\nabla_\mathbf{w}f_i(\mathbf{w}),\nabla_\mathbf{w}f_j(\mathbf{w})\rangle - \langle\nabla_{\mathbf{w}'}f_i(\mathbf{w}'),\nabla_{\mathbf{w}'}f_j(\mathbf{w}')\rangle|$$
$$= |\langle\nabla_\mathbf{w}f_i(\mathbf{w}) - \nabla_{\mathbf{w}'}f_i(\mathbf{w}') + \nabla_{\mathbf{w}'}f_i(\mathbf{w}'),\nabla_\mathbf{w}f_j(\mathbf{w})\rangle$$
$$\quad - \langle\nabla_{\mathbf{w}'}f_i(\mathbf{w}'),\nabla_{\mathbf{w}'}f_j(\mathbf{w}') - \nabla_\mathbf{w}f_j(\mathbf{w}) + \nabla_\mathbf{w}f_j(\mathbf{w})\rangle|$$
$$= |\langle\nabla_\mathbf{w}f_i(\mathbf{w}) - \nabla_{\mathbf{w}'}f_i(\mathbf{w}'),\nabla_\mathbf{w}f_j(\mathbf{w})\rangle - \langle\nabla_{\mathbf{w}'}f_i(\mathbf{w}'),\nabla_{\mathbf{w}'}f_j(\mathbf{w}') - \nabla_\mathbf{w}f_j(\mathbf{w})\rangle|$$
$$\leq |\langle\nabla_\mathbf{w}f_i(\mathbf{w}) - \nabla_{\mathbf{w}'}f_i(\mathbf{w}'),\nabla_\mathbf{w}f_j(\mathbf{w})\rangle| + |\langle\nabla_{\mathbf{w}'}f_i(\mathbf{w}'),\nabla_{\mathbf{w}'}f_j(\mathbf{w}') - \nabla_\mathbf{w}f_j(\mathbf{w})\rangle|$$
$$\leq \|\nabla_\mathbf{w}f_i(\mathbf{w}) - \nabla_{\mathbf{w}'}f_i(\mathbf{w}')\|\|\nabla_\mathbf{w}f_j(\mathbf{w})\| + \|\nabla_{\mathbf{w}'}f_i(\mathbf{w}')\|\|\nabla_{\mathbf{w}'}f_j(\mathbf{w}') - \nabla_\mathbf{w}f_j(\mathbf{w})\|$$
$$\leq L\|\mathbf{w} - \mathbf{w}'\|\|\nabla_\mathbf{w}f_j(\mathbf{w})\| + \|\nabla_{\mathbf{w}'}f_i(\mathbf{w}')\|L\|\mathbf{w}' - \mathbf{w}\|$$
$$\leq 2LM\|\mathbf{w} - \mathbf{w}'\|.$$

Here the first inequality uses the triangle inequality, the second inequality uses the Cauchy-Schwartz inequality, and the third and fourth inequalities use the assumptions that $\nabla_\mathbf{w}f_i(\mathbf{w})$ and $\nabla_\mathbf{w}f_j(\mathbf{w})$ are Lipschitz in $\mathbf{w}$ and have bounded norm. $\qquad\square$

We are now ready to prove the corollary, which we restate here. The proof uses a standard "epsilon-net" argument; we identify a fine net of points within the ball $\mathcal{B}_r$. If the gradient confusion is small at every point in this discrete set, and the gradient confusion varies slowly enough with $\mathbf{W}$, when we can guarantee small gradient confusion at every point in $\mathcal{B}_r$.

**Corollary 4.1** (Uniform concentration for all weights around the minimizer). *Select a point* $\mathbf{W} = (\mathbf{W}_0,\mathbf{W}_1,\ldots,\mathbf{W}_\beta)$, *satisfying assumption 1. Consider a ball* $\mathcal{B}_r$ *centered at* $\mathbf{W}$ *of radius* $r > 0$. *If the data* $\{\mathbf{x}_i\}_{i\in[N]}$ *are sampled uniformly from a unit sphere, then the gradient confusion bound in equation 3 holds uniformly at all points* $\mathbf{W}' \in \mathcal{B}_r$ *with probability at least*

$$1 - N^2\exp\left(-\frac{cd\eta^2}{64\zeta_0^4(\beta+2)^4}\right), \qquad\qquad \text{if } r \leq \eta/4\zeta_0^2,$$
$$1 - N^2\exp\left(-\frac{cd\eta^2}{64\zeta_0^4(\beta+2)^4} + \frac{8d\zeta_0^2r}{\eta}\right), \qquad \text{otherwise.}$$

*Proof.* Define the function $h^+(\mathbf{W}) = \max_{ij} h_{\mathbf{W}}(\mathbf{x}_i, \mathbf{x}_j)$. Our goal is to find conditions under which $h^+(\mathbf{W}) > -\eta$ for all $\mathbf{W}$ in a large set. To derive such conditions, we will need a Lipschitz constant for $h^+(\mathbf{W})$, which is no larger than the maximal Lipschitz constant of $h_{\mathbf{W}}(\mathbf{x}_i, \mathbf{x}_j)$ for all $i, j$. We have that $\|\nabla_{\mathbf{W}} f_i\| = \|\zeta_{\mathbf{x}_i}(\mathbf{W})\mathbf{x}_i\| \leq \zeta_0$. Now we need to get a $\mathbf{W}$-Lipschitz constants for $\nabla_{\mathbf{x}_i} f_i = \zeta_{\mathbf{x}_i}(\mathbf{W})\mathbf{x}_i$. By lemma D.1, we have $\|\nabla_{\mathbf{W}}(\zeta_{\mathbf{x}_i}(\mathbf{W})\mathbf{x}_i)\| = \|(\nabla_{\mathbf{W}}\zeta_{\mathbf{x}_i}(\mathbf{W}))\mathbf{x}_i\| \leq \zeta_0$. Using lemma D.2, we see that $2\zeta_0^2$ is a Lipschitz constant for $h_{\mathbf{W}}(\mathbf{x}_i, \mathbf{x}_j)$, and thus also $h^+(\mathbf{W})$.

Now, consider a minimizer $\mathbf{W}$ of the objective, and a ball $\mathcal{B}_r$ around this point of radius $r$. Define the constant $\epsilon = \frac{\eta}{4\zeta_0^2}$, and create an $\epsilon$-net of points $\mathcal{N}_\epsilon = \{\mathbf{W}_i\}$ inside the ball. This net is sufficiently dense that any $\mathbf{W}' \in \mathcal{B}_r$ is at most $\epsilon$ units away from some $\mathbf{W}_i \in \mathcal{N}_\epsilon$. Furthermore, because $h^+(\mathbf{W})$ is Lipschitz in $\mathbf{W}$, $|h^+(\mathbf{W}') - h^+(\mathbf{W}_i)| \leq 2\zeta_0^2\epsilon = \eta/2$.

We now know the following: if we can guarantee that

$$h^+(\mathbf{W}_i) \geq -\eta/2, \text{ for all } \mathbf{W}_i \in \mathcal{N}_\epsilon, \tag{10}$$

then we also know that $h^+(\mathbf{W}') \geq -\eta$ for all $\mathbf{W}' \in \mathcal{B}_r$. For this reason, we prove the result by bounding the probability that (10) holds. It is known that $\mathcal{N}_\epsilon$ can be constructed so that $|\mathcal{N}_\epsilon| \leq (2r/\epsilon + 1)^d = (8\zeta_0^2 r/\eta + 1)^d$ (see Vershynin (2018), corollary 4.1.13). Theorem 4.1 provides a bound on the probability that each individual point in the net satisfies condition (10). Using a union bound, we see that all points in the net satisfy this condition with probability at least

$$1 - N^2\left(\frac{8\zeta_0^2 r}{\eta} + 1\right)^d \exp\left(-\frac{cd(\eta/2)^2}{16\zeta_0^4}\right) \tag{11}$$

$$= 1 - N^2 \exp(d\log(8\zeta_0^2 r/\eta + 1))\exp\left(-\frac{cd\eta^2}{64\zeta_0^4}\right) \tag{12}$$

$$\geq 1 - N^2 \exp(8d\zeta_0^2 r/\eta)\exp\left(-\frac{cd\eta^2}{64\zeta_0^4}\right) \tag{13}$$

$$= 1 - N^2 \exp\left(-\frac{cd\eta^2}{64\zeta_0^4} + \frac{8d\zeta_0^2 r}{\eta}\right). \tag{14}$$

Finally, note that, if $r < \epsilon$, then we can form a net with $|\mathcal{N}_\epsilon| = 1$. In this case, the probability of satisfying (10) is at least

$$1 - N^2 \exp\left(-\frac{cd(\eta/2)^2}{64\zeta_0^4}\right). \qquad \square$$

## D.4 PROOF OF THEOREM 4.2

**Theorem 4.2** (Neural nets with randomly chosen weights). *Let $\mathbf{W}_0, \mathbf{W}_1, \ldots, \mathbf{W}_\beta$ be weight matrices chosen according to strategy 4.1. There exists fixed constants $c_1, c_2 > 0$ such that we have:*

1. *Consider a fixed but arbitrary dataset $\mathbf{x}_1, \mathbf{x}_2, \ldots, \mathbf{x}_N$ with $\|\mathbf{x}_i\| \leq 1$ for every $i \in [N]$. For $\eta > 4$, the gradient confusion bound in equation 3 holds with probability at least*

$$1 - \beta\exp\left(-c_1\kappa^2\ell^2\right) - N^2\exp\left(\frac{-c\ell^2\beta(\eta-4)^2}{64\zeta_0^4(\beta+2)^4}\right).$$

2. *If the dataset $\{\mathbf{x}_i\}_{i\in[N]}$ is such that each $\mathbf{x}_i$ is an i.i.d. sample from the surface of $d$-dimensional unit sphere, then for every $\eta > 0$ the gradient confusion bound in equation 3 holds with probability at least*

$$1 - \beta\exp\left(-c_1\kappa^2\ell^2\right) - N^2\exp\left(\frac{-c_2(\ell d+\ell^2\beta)\eta^2}{16\zeta_0^4(\beta+2)^4}\right).$$

Both parts in theorem 4.2 depend on the following argument. From theorem 2.3.8 and Proposition 2.3.10 in Tao (2012) with appropriate scaling[5], we have for every $p = 1, \ldots, \beta$ we have that the matrix norm $\|\mathbf{W}_p\| \leq 1$ with probability at least $1 - \beta\exp\left(-c_1\kappa^2\ell^2\right)$ and $\|\mathbf{W}_0\| \leq 1$ with probability at least $1 - \exp\left(-c_1\kappa^2 d^2\right)$ when the weight matrices are initialized according to strategy 4.1. Thus,

---

[5]In particular, each entry has to be scaled by $\frac{1}{\ell}$ for matrices $\{\mathbf{W}_p\}_{p\in[\beta]}$ and $\frac{1}{d}$ for the matrix $\mathbf{W}_0$.

conditioning on this event it implies that these matrices satisfy assumption 1. The proof strategy is similar to that of theorem 4.1. We will first show that the gradient of the function $h(.,.)$ as defined in equation (7) *with respect to the weights* is bounded. Note that in part (1) the random variable is the set of weight matrices $\{W_p\}_{p\in[\beta]}$. Thus, the dimension used to invoke theorem E.1 is at most $\ell^2\beta$. In part (2) along with the weights, the data $\mathbf{x}\in\mathbb{R}^d$ is also random. Thus, the dimension used to invoke theorem E.1 is at most $\ell d + \ell^2\beta$. Combining this with theorem E.1, the bound on the gradient of $h(.,.)$ and taking a union bound, we get the respective parts of the theorem. Thus, all it remains to prove is the bound on the gradient of the function $h(.,.)$ as defined in equation (7) *with respect to the weights* conditioning on the event that $\|\mathbf{W}_p\|\le 1$ for every $p\in\{0,1,\ldots,\beta\}$.

We obtain the following analogue of equation (8).

$$
(\nabla_{\mathbf{W}}\zeta_{\mathbf{x}_i}(\mathbf{W}))\zeta_{\mathbf{x}_j}(\mathbf{W})\left(\sum_{p\in[\beta]_0}\mathrm{Tr}[\mathbf{H}_{p-1}(\mathbf{x}_i)\cdot\mathbf{g}_p(\mathbf{x}_i)^\top\cdot\mathbf{g}_p(\mathbf{x}_j)\cdot\mathbf{H}_{p-1}(\mathbf{x}_i)^\top]\right)+
$$

$$
(\nabla_{\mathbf{W}}\zeta_{\mathbf{x}_j}(\mathbf{W}))\zeta_{\mathbf{x}_i}(\mathbf{W})\left(\sum_{p\in[\beta]_0}\mathrm{Tr}[\mathbf{H}_{p-1}(\mathbf{x}_i)\cdot\mathbf{g}_p(\mathbf{x}_i)^\top\cdot\mathbf{g}_p(\mathbf{x}_j)\cdot\mathbf{H}_{p-1}(\mathbf{x}_i)^\top]\right)+
$$

$$
\zeta_{\mathbf{x}_i}(\mathbf{W})\zeta_{\mathbf{x}_j}(\mathbf{W})\sum_{p\in[\beta]_0}\left[\nabla_{\mathbf{W}}\left(\mathbf{H}_{p-1}(\mathbf{x}_i)\cdot\mathbf{g}_p(\mathbf{x}_i)^\top\cdot\mathbf{g}_p(\mathbf{x}_j)\cdot\mathbf{H}_{p-1}(\mathbf{x}_i)\right)\right]^\top. \quad (15)
$$

As in the case of the proof for theorem 4.1, we will upper-bound the $\ell_2$-norm of the above expression. In particular, we show the following.

$$
\left\|(\nabla_{\mathbf{W}}\zeta_{\mathbf{x}_i}(\mathbf{W}))\zeta_{\mathbf{x}_j}(\mathbf{W})\left(\sum_{p\in[\beta]_0}\mathrm{Tr}[\mathbf{H}_{p-1}(\mathbf{x}_i)\cdot\mathbf{g}_p(\mathbf{x}_i)^\top\cdot\mathbf{g}_p(\mathbf{x}_j)\cdot\mathbf{H}_{p-1}(\mathbf{x}_i)^\top]\right)\right\|_2\le 2\zeta_0^2(\beta+2)^2.
$$
$$(16)$$

$$
\left\|(\nabla_{\mathbf{W}}\zeta_{\mathbf{x}_j}(\mathbf{W}))\zeta_{\mathbf{x}_i}(\mathbf{W})\left(\sum_{p\in[\beta]_0}\mathrm{Tr}[\mathbf{H}_{p-1}(\mathbf{x}_i)\cdot\mathbf{g}_p(\mathbf{x}_i)^\top\cdot\mathbf{g}_p(\mathbf{x}_j)\cdot\mathbf{H}_{p-1}(\mathbf{x}_i)^\top]\right)\right\|_2\le 2\zeta_0^2(\beta+2)^2.
$$
$$(17)$$

$$
\left\|\zeta_{\mathbf{x}_i}(\mathbf{W})\zeta_{\mathbf{x}_j}(\mathbf{W})\sum_{p\in[\beta]_0}\left[\nabla_{\mathbf{W}}\left(\mathbf{H}_{p-1}(\mathbf{x}_i)\cdot\mathbf{g}_p(\mathbf{x}_i)^\top\cdot\mathbf{g}_p(\mathbf{x}_j)\cdot\mathbf{H}_{p-1}(\mathbf{x}_i)\right)\right]^\top\right\|_2\le 4\zeta_0^2(\beta+2)^2.
$$
$$(18)$$

Equations (16) and 17 follow from the the fact that $\|(\nabla_{\mathbf{W}}\zeta_{\mathbf{x}_i}(\mathbf{W}))\|_2\le\zeta_0$ and the arguments in the proof for theorem 4.1. We will now show the proof sketch for equation (18). For every $p\in[\beta]_0$, consider $\|\nabla_{\mathbf{W}}\left(\mathbf{H}_{p-1}(\mathbf{x}_i)\cdot\mathbf{g}_p(\mathbf{x}_i)^\top\cdot\mathbf{g}_p(\mathbf{x}_j)\cdot\mathbf{H}_{p-1}(\mathbf{x}_i)\right)\|$. Using the symmetry between $\mathbf{x}_i$ and $\mathbf{x}_j$, the expression can be upper-bounded by,

$$
2\|\nabla_{\mathbf{W}}\mathbf{H}_{p-1}(\mathbf{x}_i)\|\|\mathbf{g}_p(\mathbf{x}_i)^\top\|\|\mathbf{g}_p(\mathbf{x}_j)\|\|\mathbf{H}_{p-1}(\mathbf{x}_i)\|+2\|\mathbf{H}_{p-1}(\mathbf{x}_i)\|\|\nabla_{\mathbf{W}}\mathbf{g}_p(\mathbf{x}_i)^\top\|\|\mathbf{g}_p(\mathbf{x}_j)\|\|\mathbf{H}_{p-1}(\mathbf{x}_i)\|.
$$

As before we can use an inductive argument to find the upper-bound and thus, we obtain the following which implies equation (18).

$$
\|\nabla_{\mathbf{W}}\left(\mathbf{H}_{p-1}(\mathbf{x}_i)\cdot\mathbf{g}_p(\mathbf{x}_i)^\top\cdot\mathbf{g}_p(\mathbf{x}_j)\cdot\mathbf{H}_{p-1}(\mathbf{x}_i)\right)\|\le 4(\beta+2)^2.
$$

Next, we show that the expected value can be lower-bounded by $-4$ as in the case of theorem 4.2 above. Combining these two gives us the desired result. Consider $\mathbb{E}_{\mathbf{W}}[h_{\mathbf{W}}(\mathbf{x}_i,\mathbf{x}_j)]$. We compute this expectation iteratively as follows.

$$
\mathbb{E}_{\mathbf{W}}[h_{\mathbf{W}}(\mathbf{x}_i,\mathbf{x}_j)]
$$
$$
=\mathbb{E}_{\mathbf{W}_0}[\mathbb{E}_{\mathbf{W}_1}[\ldots\mathbb{E}_{\mathbf{W}_\beta}[h_{\mathbf{W}}(\mathbf{x}_i,\mathbf{x}_j)]
$$
$$
\ge -4\mathbb{E}_{\mathbf{W}_0}\left[\mathbb{E}_{\mathbf{W}_1}\left[\ldots\mathbb{E}_{\mathbf{W}_\beta}\left[\sum_{p\in[\beta]_0}\mathrm{Tr}(\mathbf{H}_{p-1}(\mathbf{x}_i)\cdot\mathbf{g}_p(\mathbf{x}_i)^\top\cdot\mathbf{g}_p(\mathbf{x}_j)\cdot\mathbf{H}_{p-1}(\mathbf{x}_i)^\top)\right]\right]\right].
$$

The inequality combines Eq. (7) with Lemma D.1. We now prove the following inequality.

$$\mathbb{E}_{\mathbf{W}_0}\left[\mathbb{E}_{\mathbf{W}_1}\left[\ldots\mathbb{E}_{\mathbf{W}_\beta}\left[\sum_{p\in[\beta]_0}\mathrm{Tr}(\mathbf{H}_{p-1}(\mathbf{x}_i)\cdot\mathbf{g}_p(\mathbf{x}_i)^\top\cdot\mathbf{g}_p(\mathbf{x}_j)\cdot\mathbf{H}_{p-1}(\mathbf{x}_i)^\top)\right]\right]\right]\leq 1. \quad (19)$$

Consider the inner-most expectation. Note that the only random variable is $\mathbf{W}_\beta$. Moreover, the term inside the trace is *scalar*. Note that the activation function $\sigma$ satisfies $|\sigma'(x)|\leq 1$. Using the linearity of expectation, the LHS in equation (19) can be upper-bounded by the following.

$$\mathbb{E}_{\mathbf{W}_0}\left[\mathbb{E}_{\mathbf{W}_1}\left[\ldots\mathbb{E}_{\mathbf{W}_{\beta-1}}\left[\mathrm{Tr}(\mathbf{H}_{\beta-1}(\mathbf{x}_i)\cdot\mathbf{H}_{\beta-1}(\mathbf{x}_i)^\top)\right]\right]\right] \quad (20)$$

$$+\mathbb{E}_{\mathbf{W}_0}\left[\mathbb{E}_{\mathbf{W}_1}\left[\ldots\mathbb{E}_{\mathbf{W}_\beta}\left[\sum_{p\in[\beta]_0\setminus\{\beta\}}\mathrm{Tr}(\mathbf{H}_{p-1}(\mathbf{x}_i)\cdot\mathbf{g}_p(\mathbf{x}_i)^\top\cdot\mathbf{g}_p(\mathbf{x}_j)\cdot\mathbf{H}_{p-1}(\mathbf{x}_i)^\top)\right]\right]\right]. \quad (21)$$

The first sum in the above expression can be upper-bounded by 1, since $|\sigma(x)|\leq 1$. We will now show that the second sum is 0. Consider the inner-most expectation. The weights $\mathbf{W}_\beta$ appears only in the expression $\mathbf{g}_p(\mathbf{x}_i)^\top\cdot\mathbf{g}_p(\mathbf{x}_j)$. Moreover, note that every entry in $\mathbf{W}_\beta$ is an i.i.d. normal random variable with mean 0. Thus, the second summand simplifies to,

$$\mathbb{E}_{\mathbf{W}_0}\left[\mathbb{E}_{\mathbf{W}_1}\left[\ldots\mathbb{E}_{\mathbf{W}_{\beta-1}}\left[\sum_{p\in[\beta]_0\setminus\{\beta,\beta-1\}}\mathrm{Tr}(\mathbf{H}_{p-1}(\mathbf{x}_i)\cdot\mathbf{g}_p(\mathbf{x}_i)^\top\cdot\mathbf{g}_p(\mathbf{x}_j)\cdot\mathbf{H}_{p-1}(\mathbf{x}_i)^\top)\right]\right]\right].$$

Applying the above argument repeatedly we obtain that the second summand (equation (21)) is 0.

Thus, we obtain the inequality in equation (19) which implies that $\mathbb{E}_{\mathbf{W}}[h_{\mathbf{W}}(\mathbf{x}_i,\mathbf{x}_j)]\geq -4$.

## E  TECHNICAL LEMMAS

We will briefly describe some technical lemmas we require in our analysis. The following Chernoff-style concentration bound is proved in Chapter 5 of Vershynin (2018).

**Lemma E.1** (Concentration of Lipshitz function over a sphere). *Let* $\mathbf{x}\in\mathbb{R}^d$ *be sampled uniformly from the surface of a $d$-dimensional sphere. Consider a Lipshitz function $\ell:\mathbb{R}^d\to\mathbb{R}$ which is differentiable everywhere. Let $\|\nabla\ell\|_2$ denote $\sup_{\mathbf{x}\in\mathbb{R}^d}\|\nabla\ell(\mathbf{x})\|_2$. Then for any $t\geq 0$ and some fixed constant $c\geq 0$, we have the following.*

$$\Pr\left[\left|\ell(\mathbf{x})-\mathbb{E}[\ell(\mathbf{x})]\right|\geq t\right]\leq 2\exp\left(-\frac{cdt^2}{\rho^2}\right), \quad (22)$$

*where* $\rho\geq\|\nabla\ell\|_2$.

We will rely on the following generalization of lemma E.1. We would like to point out that the underlying metric is the Euclidean metric and thus we use the $\|.\|_2$-norm.

**Corollary E.1.** *Let* $\mathbf{x},\mathbf{y}\in\mathbb{R}^d$ *be two mutually independent vectors sampled uniformly from the surface of a $d$-dimensional sphere. Consider a Lipshitz function $\ell:\mathbb{R}^d\times\mathbb{R}^d\to\mathbb{R}$ which is differentiable everywhere. Let $\|\nabla\ell\|_2$ denote $\sup_{(\mathbf{x},\mathbf{y})\in\mathbb{R}^d\times\mathbb{R}^d}\|\nabla\ell(\mathbf{x},\mathbf{y})\|_2$. Then for any $t\geq 0$ and some fixed constant $c\geq 0$, we have the following.*

$$\Pr\left[\left|\ell(\mathbf{x},\mathbf{y})-\mathbb{E}[\ell(\mathbf{x},\mathbf{y})]\right|\geq t\right]\leq 2\exp\left(-\frac{cdt^2}{\rho^2}\right), \quad (23)$$

*where* $\rho\geq\|\nabla\ell\|_2$.

*Proof.* This corollary can be derived from lemma E.1 as follows. Note that for every fixed $\tilde{\mathbf{y}}\in\mathbb{R}^d$, equation 22 holds. Additionally, we have that the vectors $\mathbf{x}$ and $\mathbf{y}$ are mutually independent. Hence we can write the LHS of equation 23 as the following.

$$\int_{(\tilde{\mathbf{y}})_1=-\infty}^{(\tilde{\mathbf{y}})_1=\infty}\ldots\int_{(\tilde{\mathbf{y}})_d=-\infty}^{(\tilde{\mathbf{y}})_d=\infty}\Pr\left[\left|\ell(\mathbf{x},\mathbf{y})-\mathbb{E}[\ell(\mathbf{x},\mathbf{y})]\right|\geq t\;\middle|\;\mathbf{y}=\tilde{\mathbf{y}}\right]\phi(\tilde{\mathbf{y}})d(\tilde{\mathbf{y}})_1\ldots d(\tilde{\mathbf{y}})_d.$$

Here $\phi(\tilde{\mathbf{y}})$ refers to the pdf of the distribution of $\mathbf{y}$. From independence, the inner term in the integral evaluates to $\Pr\left[\left|\ell(\mathbf{x}, \tilde{\mathbf{y}}) - \mathbb{E}[\ell(\mathbf{x}, \tilde{\mathbf{y}})]\right| \geq t\right]$. We know this is less than or equal to $2\exp\left(-\frac{cdt^2}{\|\nabla \ell\|_2^2}\right)$. Therefore, the integral can be upper bounded by the following.

$$\int_{(\tilde{\mathbf{y}})_1=-\infty}^{(\tilde{\mathbf{y}})_1=\infty} \cdots \int_{(\tilde{\mathbf{y}})_d=-\infty}^{(\tilde{\mathbf{y}})_d=\infty} 2\exp\left(-\frac{cdt^2}{\|\nabla \ell\|_2^2}\right) \phi(\tilde{\mathbf{y}}) d(\tilde{\mathbf{y}})_1 \ldots d(\tilde{\mathbf{y}})_d.$$

Since $\phi(\tilde{\mathbf{y}})$ is a valid pdf, we get the required equation 23. □

Additionally, we will use the following facts about a normalized Gaussian random variable.

**Lemma E.2.** *For a normalized Gaussian* $\mathbf{x}$ *(i.e., an* $\mathbf{x}$ *sampled uniformly from the surface of a unit d-dimensional sphere) the following statements are true.*

1. $\forall p \in [d]$ *we have that* $\mathbb{E}[(\mathbf{x})_p] = 0$.

2. $\forall p \in [d]$ *we have that* $\mathbb{E}[(\mathbf{x})_p^2] = 1/d$.

*Proof.* Part (1) can be proved by observing that the normalized Gaussian random variable is spherically symmetric about the origin. In other words, for every $p \in [d]$ the vectors $(x_1, x_2, \ldots, x_p, \ldots, x_d)$ and $(x_1, x_2, \ldots, -x_p, \ldots, x_d)$ are identically distributed. Hence $\mathbb{E}[x_p] = \mathbb{E}[-x_p]$ which implies that $\mathbb{E}[x_p] = 0$.

Part (2) can be proved by observing that for any $p, p' \in [d]$, $x_p$ and $x_{p'}$ are identically distributed. Fix any $p \in [d]$. We have that $\sum_{p' \in [d]} \mathbb{E}[x_{p'}^2] = d \times \mathbb{E}[x_p^2]$. Note that we have

$$\sum_{p' \in [d]} \mathbb{E}[x_{p'}^2] = \int_{(\mathbf{x})_1=-\infty}^{(\mathbf{x})_1=\infty} \cdots \int_{(\mathbf{x})_d=-\infty}^{(\mathbf{x})_d=\infty} \frac{\sum_{p' \in [d]} x_{p'}^2}{\sum_{p'' \in [d]} x_{p''}^2} \phi(\mathbf{x}) d(\mathbf{x})_1 \ldots d(\mathbf{x})_d = 1.$$

Therefore $\mathbb{E}[x_p^2] = 1/d$. □

We use the following well-known Gaussian concentration inequality in our proofs (*e.g.,* Chapter 5 in Boucheron et al. (2013)).

**Lemma E.3** (Gaussian Concentration). *Let* $\mathbf{x} = (x_1, x_2, \ldots, x_d)$ *be i.i.d.* $\mathcal{N}(0, \nu^2)$ *random variables. Consider a Lipshitz function* $\ell : \mathbb{R}^d \to \mathbb{R}$ *which is differentiable everywhere. Let* $\|\nabla \ell\|_2$ *denote* $\sup_{\mathbf{x} \in \mathbb{R}^d} \|\nabla \ell(\mathbf{x})\|_2$. *Then for any* $t \geq 0$, *we have the following.*

$$\Pr\left[\left|\ell(\mathbf{x}) - \mathbb{E}[\ell(\mathbf{x})]\right| \geq t\right] \leq 2\exp\left(-\frac{t^2}{2\nu^2\rho^2}\right), \tag{24}$$

*where* $\rho \geq \|\nabla \ell\|_2$.

## F    ADDITIONAL DISCUSSION OF THE SMALL WEIGHTS ASSUMPTION (ASSUMPTION 1)

Without the small-weights assumption, the signal propagated forward or the gradients $\nabla_{\mathbf{W}} f_i$ could potentially blow up in magnitude, making the network untrainable. Proving non-vacuous bounds in case of such blow-ups in magnitude of the signal or the gradient is not possible in general, and thus, we assume this restricted class of weights.

Note that the small-weights assumption is not just a theoretical concern, but also usually enforced in practice. Neural networks are often trained with *weight decay* regularizers of the form $\sum_i \|W_i\|_F^2$, which keep the weights small during optimization. The operator norm of convolutional layers have also recently been used as an effective regularizer as well for image classification tasks by Sedghi et al. (2018).

In the proof of theorem 4.2 we showed that assumption 1 holds at initialization with high probability. While, in general, there is no reason to believe that such a small-weights assumption would continue to hold during optimization without explicit regularizers like weight decay, some recent work has shown evidence that the weights do not move too far away during training from the random initialization point for overparameterized neural nets (Neyshabur et al., 2018; Dziugaite & Roy, 2017; Nagarajan & Kolter, 2019; Zou et al., 2018; Allen-Zhu et al., 2018; Du et al., 2018; Oymak & Soltanolkotabi, 2018). It is worth noting though that all these results have been shown under some restrictive assumptions, such as the width requiring to be much larger than generally used by practitioners.

