# OpenReview forum: "The Effect of Neural Net Architecture on Gradient Confusion & Training Performance"
_ICLR.cc/2020/Conference — Reject_

### Official Review · AnonReviewer2 · 2019-10-20
**Official Blind Review #2**

**Rating:** 1

**Review:**

This paper targets to understand some key factors that might influence the training speed of neural networks. It defines a concept called "gradient confusion" to quantify these factors. Roughly speaking, this term captures the disagreement of the descending directions suggested by different samples.

On the positive side, to understand the training procedure is important to the deep learning community. The idea on "gradient confusion" is quite straightforward and easy to understand. The paper is clearly written.

However, there are some essential drawbacks of the paper that make me lean towards rejecting it.

First, this concept of "gradient confusion" is very similar to the "gradient diversity" introduced more than 2 years ago in [1]. The term proposed in [1] measures the "gradient confusion" relative to the average of gradient norms, and is an averaged case rather than the worst case version. (This actually brings a second issue of the "gradient confusion" which makes the definition less useful.) However the paper has not discussed this existing work and the contribution on top of it.

Second, the "gradient confusion" is not a robust term even to one outlier sample. Since the definition in Eqn. (3) measures the worst case scenario, one could add an outlier sample that easily makes the eta arbitrarily large and the latter bound on the convergence will be meaningless. In comparison, the "gradient diversity" in [1] of the averaged value makes more sense to me.

Third, the proposal of a new theory should be in favor of at least some applicable cases. In this paper, the take-home message seems to be when the samples are pushing the gradient towards the same direction, the training becomes faster, which is very legit and people believe this. However, what can we do about this? The authors fail to propose some interesting applications that could make use of the "gradient confusion" to help the training. For instance, the work of [1] has made use of the gradient diversity to accelerate distributed learning which could be one application. I encourage the authors to work towards some useful applications concerning this new concept.

To sum up, I don't think this paper brings out some real contributions to the community and should not be accepted.


[1]  Yin, Dong, Ashwin Pananjady, Max Lam, Dimitris Papailiopoulos, Kannan Ramchandran, and Peter Bartlett. "Gradient diversity: a key ingredient for scalable distributed learning." arXiv preprint arXiv:1706.05699 (2017).

**Experience Assessment:**

I have published in this field for several years.

**Review Assessment: Checking Correctness Of Derivations And Theory:**

I assessed the sensibility of the derivations and theory.

**Review Assessment: Checking Correctness Of Experiments:**

I carefully checked the experiments.

**Review Assessment: Thoroughness In Paper Reading:**

I read the paper at least twice and used my best judgement in assessing the paper.

---

> ### Author Response · Authors · 2019-11-12
> **Response to review (1/2)**
>
> We thank the reviewer for their assessment of our work. Unfortunately, we think the reviewer had some misunderstandings about the connection with our paper to prior work, which we discuss below. We hope the reviewer might reconsider their score given our clarifications.
>
> 1. Firstly, note that we do discuss the connections between gradient confusion and gradient diversity. We do this in appendix H (page 29), and point to this section using footnote 2 on page 3 in the main paper. We can certainly move some of this discussion into the main paper.
>
> 2. Secondly, note that gradient diversity is *not* the average case of gradient confusion. There are crucial differences between the two metrics (which we elaborate on in appendix H, and mention again here in this response), and these differences affect the conclusions on studying one property vs. the other.
> Gradient diversity, similar to gradient confusion, also measures the degree to which individual gradients at different data samples are different from each other. However, note that the gradient diversity measure gets larger as the individual gradients become orthogonal to each other, and further increases as the gradients start pointing in opposite directions. In a large batch, higher gradient diversity is desirable, and this leads to improved convergence rates in distributed settings, as shown in Yin et al. (2017).
> On the other hand, gradient confusion between two individual gradients is zero unless the inner product between them is negative. This makes gradient confusion useful for studying convergence of small minibatch SGD. This is because different possible SGD updates do not conflict with each other unless they are negatively correlated with each other..
> The choice of the definition of gradient diversity in Yin et al. (2017) has important implications when its behavior is studied in over-parameterized settings. Chen et al. (2018) extends the work of Yin et al. (2017), where the authors prove on 2-layer neural nets (and multi-layer linear neural nets) that gradient diversity *increases* with increased width and decreased depth. This metric does not however distinguish between the cases where gradients become more orthogonal vs. more negatively correlated. As we show in our paper, we show that this can have different effects on the convergence of SGD in overparameterized settings. Specifically, we show that increased width and decreased depth *decreases* gradient confusion, and makes these networks easier to train. In fact, we see that the gradients become more orthogonal to each other in this case (see for example the right plots in figures 2 and 3). Thus, we view our papers to be complementary to each other, providing insights about different issues (large batch distributed training vs. small minibatch convergence).
> Finally, we wanted to point out that our theoretical results hold for very general neural nets (for arbitrary depth neural nets with all popular non-linearities and loss functions). In contrast, the gradient diversity papers consider 2 layer neural nets and multilayer *linear* nets and the squared loss function for their theoretical results.
>
> 3. Further note that all the results in our paper can be trivially extended to using a bound on the average gradient inner product: $\sum_{i, j = 1}^N \langle \nabla f_i(\mathbf{w}), \nabla f_j (\mathbf{w})  \rangle / N^2  \geq -\eta$. All the results would remain the same up to constants. Note we are studying over-parameterized problems, so the infinite data case ($N \rightarrow \infty$) is not valid. We will clarify this in the paper.

---

> > ### Author Response · Authors · 2019-11-12
> > **Response to review (2/2)**
> >
> > Continuing the above response below.
> >
> > 4. Lastly, we wanted to point out that our work has a number of relevant practical insights. Our proposed metric, gradient confusion, is a new measure to analyze performance on neural nets. We provide novel and tight bounds on the relationship between network depth, layer width, input dimension and SGD performance. Further, through extensive empirical results, we analyze the effect of batch normalization and skip connections on gradient confusion. Thus our results provide a number of new and important insights that can be used for better neural net model design, as well as for better algorithms for better training and generalization.
> > In fact, since our paper has gone up on arXiv, a number of papers have already used these insights from our paper for a number of practically relevant contributions:
> > - The same metric as gradient confusion was used recently for understanding and analyzing generalization performance on neural nets.
> > - Insights from the paper on the effect of batch normalization on gradient confusion was used for both understanding the different properties of batch normalization, as well as to come up with simple replacements for batch normalization.
> > - The gradient confusion measure has also been directly used as a metric in neural architecture search algorithms. These algorithms were shown to be very effective at finding model architectures that are easy to train and attain state-of-the-art performance on a number of standard benchmarks.
> > Unfortunately, to preserve anonymity, we cannot directly cite these papers in our response. Further, we are aware of some of the insights in our work being currently used to explore its implications for adversarial training. As all of these efforts show, our results are highly general and applicable in a very broad range of problems.

---

### Official Review · AnonReviewer3 · 2019-10-23
**Official Blind Review #3**

**Rating:** 8

**Review:**


[Summary]
This paper introduces gradient confusion, a bound on the negated dot product of gradients at two data points, and studies its effect on the optimization of neural networks with SGD. If gradient confusion is high, reducing the loss on one data point with SGD increases it on another data point. Theoretical results show that (1) with a fixed learning rate, lower gradient confusion results in faster learning, and (2) increasing the width and decreasing the depth of a network reduces gradient confusion. The experiments corroborate these findings and also show that batch normalization and skip connections can reduce gradient confusion and speed up the training.

[Decision]
I vote for accepting this paper. Although most of the results mirror classical bounds that considered gradient variance, the introduced measure allows analyzing the speed of learning even if gradient variance itself is unbounded or hard to analyze. This can help with understanding the current architectures and designing new ones with desirable properties.

[Comments]
Theorem 3.1 shows that, for a constant learning rate, gradient confusion controls the SGD noise floor. If the noise floor is small, convergence to a low error is possible with a larger learning rate, and this seems to be why reducing gradient confusion speeds up SGD. In the experiments, I would expect that the chosen learning rate for networks with low gradient confusion would generally be higher. Reporting the best learning rate schedule for each network can clarify if this is true.

Are the effects of width, depth, batch normalization, and skip connections on L and \mu in Theorem 3.1 well understood? It is possible that these architectural choices change the speed of learning by modifying these constants rather than just reducing gradient confusion.

In section G in the Appendix it is said: "In general, it is not tractable to prove the concentration results in section 4 using the covariance matrix of the gradients alone without further unrealistic assumptions...." Why is bounding the gradient variance under the assumptions like bounded weights and standard initialization hard (or impossible) and a different measure like gradient confusion is necessary?

[Minor remarks]
- In Section 5: "selected the run that achieved the lowest training loss value"->"selected the learning rate that achieved..."
-------------
After rebuttal: I have read the author's response, the other reviews, and the modification. The response addresses my questions. The new revision shows the proposed measure's practical significance and robustness to one sample. I raised my score.

**Experience Assessment:**

I have read many papers in this area.

**Review Assessment: Checking Correctness Of Derivations And Theory:**

I assessed the sensibility of the derivations and theory.

**Review Assessment: Checking Correctness Of Experiments:**

I carefully checked the experiments.

**Review Assessment: Thoroughness In Paper Reading:**

I read the paper at least twice and used my best judgement in assessing the paper.

---

> ### Author Response · Authors · 2019-11-12
> **Response to review**
>
> We thank the reviewer for their positive comments. We respond to each of the reviewer’s comments below.
>
> 1. Indeed it seems to be the case that the optimal learning rates for shallower networks (which have lower gradient confusion) are higher than deeper networks (which have higher gradient confusion). For example, if we consider the CNN-$\beta$-2 results presented in figure 2 in the paper, the optimal initial learning rates are as follows:
> $\alpha_{10} = 0.3$, $\alpha_{16} = 0.1$, $\alpha_{22} = 0.03$, $\alpha_{28} = 0.03$, $\alpha_{34} = 0.01$ and $\alpha_{40} = 0.003$, where the subscript on $\alpha$ denotes the depth of the network $\beta$. Thus, the optimal learning rate decreases as the depth increases. We found this property to hold across all of our experiments where we vary the network depth. We found this property to also hold if we consider batch normalized residual nets compared to networks without batch normalization or skip connections. This property does not seem to hold consistently however for the case of changing width. We will clarify these results in the paper.
>
> 2. The width, depth, batch normalization and skip connections would certainly have an effect on the conditioning of the loss landscape (and thus affect $L$ and $\mu$). For example, previous papers have shown that batch normalization improves the conditioning of the loss landscape [1, 2]. Further, as we mention in the paper, a number of recent papers indicate that the loss surface simplifies considerably when the layers are very wide. That being said, note that the gradient confusion is a measure of the noise across data samples, and is related to the gradient variance. A number of recent papers [3, 4] have established that minibatch optimization on deep nets typically experience two different regimes:
> - when the batch size is small, optimization dynamics is primarily governed by the noise in the gradients,
> - when the batch size is large, optimization dynamics is primarily governed by the curvature of the loss landscape.
> This is the reason (i.e., to isolate the effect of gradient confusion), we consider small minibatch optimization in this paper, where the dynamics is governed primarily by the noise in the gradients and curvature would not play a significant role. Further note that we confirm this in our experiments by plotting the gradient confusion for each of the networks considered, and we see a clear correlation between higher gradient confusion corresponding to harder to train networks.
>
> 3. We think that understanding the effect of the network depth and the layer width on the gradient variance is a harder problem compared to analyzing the effect on the gradient confusion. The gradient confusion metric helps us leverage tight concentration bounds from high dimensional probability to prove new tight results that are applicable to very general neural nets with minimal assumptions (our results hold on networks with arbitrary depth and with standard non-linearities and loss functions). We are not aware of similar tight bounds for the covariance matrix without introducing additional simplifying assumptions.
>
> [1] Behrooz Ghorbani, Shankar Krishnan, and Ying Xiao. "An investigation into neural net optimization via hessian eigenvalue density." ICML 2019.
> [2] Shibani Santurkar, Dimitris Tsipras, Andrew Ilyas, and Aleksander Madry. "How does batch normalization help optimization?." NeurIPS 2018.
> [3] Guodong Zhang, Lala Li, Zachary Nado, James Martens, Sushant Sachdeva, George E. Dahl, Christopher J. Shallue, and Roger Grosse. "Which algorithmic choices matter at which batch sizes? insights from a noisy quadratic model." NeurIPS 2019.
> [4] Anonymous submission to ICLR 2020. “Hyperparameter Tuning and Implicit Regularization in Minibatch SGD”

---

### Official Review · AnonReviewer1 · 2019-10-24
**Official Blind Review #1**

**Rating:** 3

**Review:**

This paper introduces the concept of "gradient confusion" to explain why neural networks train fast with SGD. They also study the effects of width, depth on gradient confusion.
- The theoretical results assume that the data is sampled from a sphere and do not really give much insight into the effect of width and depth.
- There are some confounding factors in the experiments and there needs to be a better comparison to some related work.
Detailed review below:
Section 1:
- Please clarify how "gradient confusion" relate to the interpolation conditon of Ma et al, 2017 and the strong growth condition of Vaswani et al?
- If we run SGD with a constant step-size, it will bounce around the optimal point in a ball with radius that depends on the step-size. If I keep decreasing the step-size, this radius shrinks. How does gradient confusion relate to the step-size? Is it upper-bounded by a quantity that depends on the step-size and the batch-size?
Section 2:
- Definition 2.1: Why should this condition hold for "all" points w? Isn't it necessary only at w^* or in a small neighborhood around it?
- The gradient confusion parameter \eta should depend on the batch-size. Please clarify this.
- Figure 1: Previous work (Ma et al, Vaswani et al, Gunasekhar, 2017) all have shown that fast convergence can be obtained using SGD with a constant step-size with over-parametrized models and explained it using interpolation. What is the additional insight from gradient confusion?
- "Suppose that there is a Lipschitz constant for the Hessian" - This is a strong assumption and a vague argument, that is confusing rather than insightful. Please justify why this is a valid assumption for neural network models.
Section 3:
- If E_i || \nabla f_i(w)  ||^2 = O(\epsilon) => gradient confusion = O(\epsilon). Isn't the E_i || \nabla f_i(w)  ||^2 exactly the strong growth condition in Vaswani, et al. Can the gradient confusion results be directly derived from the results in that paper? Please compare. Also compare and cite "Stochastic Approximation of Smooth and Strongly Convex Functions:
Beyond the O(1/T ) Convergence Rate", COLT 2019.
Section 4:
- Please compare against the previous results that assumed the data to be sampled from a sphere.
- Thm 4.1: The theorem bounds the probability that gradient confusion holds for a given \eta. But the bounds of section 3 are vacuous even if the theorem holds with probability one, but for a large value of \eta. There needs to be an upper bound on \eta. Please clarify this.
- Please compare against the results of this paper by Arora et al: "On the Optimization of Deep Networks: Implicit Acceleration by Overparameterization"
- For the effect of layer width, the analysis is only for the initializated weights and does not consider the optimization, which is what the paper claimed in the introduction. Am I missing something? Please justify this. The gradient confusion can decrease as the optimization progresses?
Section 5:
- "We reduce the learning rate by a factor of 10" But all the theory is for a constant step-size. Please explain this discrepancy.
- in Figure 2, in the second figure, why is there a sharp full in the pairwise cosine similarities.
- In all these experiments, explain why the batch-size and the step-size is not a confounding factor?

**Experience Assessment:**

I have published one or two papers in this area.

**Review Assessment: Checking Correctness Of Derivations And Theory:**

I assessed the sensibility of the derivations and theory.

**Review Assessment: Checking Correctness Of Experiments:**

I carefully checked the experiments.

**Review Assessment: Thoroughness In Paper Reading:**

I read the paper thoroughly.

---

> ### Author Response · Authors · 2019-11-11
> **Response to review (1/2)**
>
> We thank the reviewer for their detailed comments about our work. We respond to each of the points brought up in their review below. As the reviewer will notice, we already discussed many of the questions in various appendix sections, and we clarify the rest of the reviewer’s questions in this response. We hope the reviewer might reconsider their score given our clarifications.
>
> 1. “The theoretical results assume that the data is sampled from a sphere and do not really give much insight into the effect of width and depth.”:
> Theorem 4.2 part 1 in the paper specifically considered the case where data is not sampled from a sphere, and instead is arbitrary but bounded. We consider the results presented in section 4 to be novel and tight, directly analyzing the effect of depth and width, and without requiring restrictive assumptions like very wide networks that is common in other recent work. The theoretical results presented are clearly practically relevant as well, as we verify through our extensive experiments (see section 5 and appendix A).
>
> 2. Relation to the interpolation and strong growth condition:
> Note that we briefly discussed this in the second paragraph of section 3 (page 4), and elaborated on the connections between the gradient variance and gradient confusion further in appendix G (page 28). The interpolation condition and the strong growth condition imply that the gradient variance at the minimizer would be small. In the aforementioned sections, we elaborated on how a small gradient variance implies small gradient confusion, but not the other way around, and discussed how this affects convergence of SGD. Further, as we mentioned in appendix G, it is not tractable to prove the concentration results in section 4 using the covariance matrix of the gradients alone without further unrealistic assumptions, such as very wide networks. A key contribution of this paper is to identify a suitable surrogate (i.e., the gradient confusion bound) to help us study the relationship between depth, width and training performance using new tight and clean bounds.
>
> 3. Relation of the gradient confusion to the step size:
> This is an important point that we elaborated on in appendix G. Neither the gradient variance nor the gradient confusion depend directly on the step size. However, as the reviewer correctly points out, when the gradient variance is bounded, the variance of the SGD updates can be decreased by decreasing the step size. Bounded gradient confusion does not however imply bounded gradient variance, and in this case decreasing the step size does not necessarily decrease the variance of the updates.
>
> 4. “Why should this condition hold for all points w”:
> The gradient confusion definition holds at a fixed weight w. The convergence rate results in section 3 requires the gradient confusion bound to hold at every point along SGD’s path. While it is possible this can be improved (we leave exploring this for future work), we think this is because bounded gradient confusion does not necessarily imply bounded gradient variance. When the gradient variance is bounded (as in most practical scenarios), the convergence rate results would require the gradient confusion bound to hold only at the minimizer. We discussed this further in appendix G.
>
> 5. Dependence on batch size:
> The expected value of the inner product between two minibatch gradients uniformly sampled from the dataset is the squared norm of the true gradient. The variance of the inner product of two minibatch gradients. on the other hand, scales down as 1/B^2, where B is the batch size. We will clarify this in the paper.
>
> 6. “Lipschitz constant on the Hessian”:
> There’s lots of work on second order optimization that establishes that the curvature matrix changes slowly for deep learning optimization. See for example “Second-Order Optimization for Neural Networks” by Martens. A Lipschitz constant for the Hessian is also a fairly standard assumption in the optimization literature, and appears in Nesterov’s classic book “Introductory Lectures on Convex Optimization”. That being said, we agree with the reviewer that this paragraph is definitely not very rigorous and a bit informal, as we mentioned in the paper. We can clarify this further.
>
> 7. Upper bound on eta:
> Yes the gradient confusion needs to be bounded for SGD to converge. We will elaborate on this in the main text.
>
> 8. Arora et al’s paper:
> We compared with Arora et al.’s very nice paper in appendix H. We will also add a discussion of Zhang & Zhou’s paper in this section. Thanks for pointing us to their paper.

---

> > ### Author Response · Authors · 2019-11-11
> > **Response to review (2/2)**
> >
> > 9. Width result at initialization:
> > Yes our analysis on the effect of width applies at standard neural net initializations. We will clarify this in the introduction. Note that for arbitrary bounded weight matrices, the output of a neural net might have no dependence on the width. A simple example to show this is to consider the case  where each weight matrix in the neural network has exactly one non-zero element, which is set to 1. The operator norm of each such weight matrix is 1, but the forward or backward propagated signals would not depend on the width.
> > For this reason, almost all recent theoretical results on neural nets have analyzed the effect of width at initialization. Further, it is possible to prove that when the layers are very wide (arguably unrealistically wide), the weights move very little, and most of the conditions at initialization persist during training. Under this assumption, several authors have analyzed the effect of width (Jacot et al., 2018; Lee et al., 2019; Lee et al., 2019; Du et al., 2018; Allen-Zhu et al., 2018; among others). Using a standard application of the techniques used in these papers, our results can be easily extended to hold during optimization on very wide networks. We will add a more detailed discussion about this in the paper.
> >
> > 10. Batch size and learning rates in experiments:
> > Note that for making sure our theoretical results were of practical relevance, we kept the standard experimental setup that’s used by practitioners when training CIFAR-10 models. This included using a step size decay, where we use the same schedule used in the original papers for these models.
> > However, to also test that our theory holds for constant step sizes, we do extensive experiments with constant step sizes in appendix A.1. Through our many experiments on a range of models and datasets, we find our theory holds remarkably well on constant step sizes, as well as on popular step size schedules used by practitioners on modern overparameterized neural nets.
> > Further, note that we use the same minibatch size for calculating gradient confusion as we use for training as this is the relevant quantity for SGD while training. Further note that we tune the step size separately for every experiment we run using a comprehensive grid search. Thus, given a problem and a batch size, we are always presenting results for the proper step size avoiding confounding factors.

---

### Author Response · Authors · 2019-11-14
**Updated version of the paper addressing all reviewer comments**

Dear AC and all reviewers,

Thank you for all your comments. We have uploaded a new version of the paper taking into account all the comments from the reviewers.

The major points brought up in the reviews are as follows:

1. Connections to prior work:
As we mention in our responses to each of the reviewers, we had discussed most of the relevant prior work and connections to in an appendix section of the paper. In the updated version of the paper, we have now moved this to the main paper, as well as added additional discussion.

2. Significance of the results:
As we mention in our responses, our theoretical results are very general, considering a wide range of different assumptions, and hold for a wide range of standard neural nets. Further, through our extensive experimental sections (section 5 and appendix A), we have shown that our theoretical results hold remarkably well across a wide range of standard models and datasets. Further, as we mention in our response, our results provide a number of new and important insights that can be used for better neural net model design, as well as for better algorithms for better training and generalization.

We believe we have addressed all the comments in our responses and with our updated version of the paper. More specifically, in the new version of the paper, we make the following changes:

1. We have moved a lot of the discussion on connections to related work from appendix sections G and H into the main paper in a new section titled “Connections to related work” (section 6). Here we address connections to the gradient variance and the interpolation and strong growth conditions, connections to the gradient diversity condition, as well as other related work.

2. We have added clarifications in section 2 of the paper on the effect of batch size on gradient confusion, and on the fact that our results remain unchanged when considering the average case definition of gradient confusion.

3. We have added clarifications on our informal discussion on low-rank Hessians in section 2.

4. We have added a slightly stronger version of theorem 4.2, which makes the effect of width on trainability of neural nets clearer. We have also added a discussion of the effect of width at initialization vs. during training in section 4, and clarified our theoretical contributions in the introduction.

5. We have added a clarification on the upper bound of eta in section 3.

6. We have expanded on the significance of the work and the implications of our results in the conclusions section (section 7).

In light of our changes, we hope the reviewers might reconsider their scores.

---

### Decision · Program_Chairs · 2019-12-19

**Decision:**

Reject

**Comment:**

This paper introduces the concept of gradient confusion to show how the neural network architecture affects the speed of training. The reviewers' opinion on this paper varies widely, also after the discussion phase.  The main disagreement is on the significance of this work, and whether the concept of gradient confusion adds something meaningful to the existing literature with respect to understanding deep networks. The strong disagreement on this paper suggest that the paper is not quite ready yet for ICLR, but that the authors should make another iteration on the paper to strengthen the case for its significance.